



# Reconciling the bottom-up and top-down estimates of the methane chemical sink using multiple observations

Yuanhong Zhao[1,2], Marielle Saunois[2], Philippe Bousquet[2], Xin Lin[2], Michaela I. Hegglin[3, 4], Josep G. Canadell[5], Robert B. Jackson[6], and Bo Zheng[7]

[1]College of Oceanic and Atmospheric Sciences, Ocean University of China, Qingdao, 266100, People's Republic of China

[2] Laboratoire des Sciences du Climat et de l'Environnement, LSCE-IPSL (CEA-CNRS-UVSQ), Université Paris-Saclay, 91191 Gif-sur-Yvette, France

[3] Institute of Energy and Climate Research – Stratosphere (IEK-7), Forschungszentrum Jülich GmbH, 52425 Jülich, Germany.

[4] Department of Meteorology, University of Reading, Earley Gate, Reading RG6 6BB, United Kingdom

[5] Global Carbon Project, CSIRO Oceans and Atmosphere, Canberra, Australian Capital Territory 2601, Australia

[6] Earth System Science Department, Woods Institute for the Environment, and Precourt Institute for Energy, Stanford University, Stanford, CA 94305, USA

[7] Institute of Environment and Ecology, Tsinghua Shenzhen International Graduate School, Tsinghua University, Shenzhen 518055, China.

*Correspondence to*: Yuanhong Zhao (zhaoyuanhong@ouc.edu.cn)





## Abstract

The methane chemical sink estimated by atmospheric chemistry models (bottom-up method) is significantly larger than estimates based on methyl-chloroform (MCF) inversions (top-down method). The difference is partly attributable to large uncertainties in hydroxyl radical (OH) concentrations simulated by the atmospheric chemistry models used to derive the bottom-up estimates. In this study, we propose a new approach based on OH precursor observations and a chemical box model. This approach improves the 3-dimensional distributions of tropospheric OH radicals obtained from atmospheric chemistry models and reconciles the bottom-up and top-down estimates of the methane sink due to chemical loss. By constraining the model simulated OH precursors with observations, the global tropospheric mean OH concentration ($[OH]_{trop-M}$) is ~$10\times10^5$ molec cm$^{-3}$ (which is $2\times10^5$ molec cm$^{-3}$ lower than the original model-simulated global $[OH]_{trop-M}$) and agrees with that obtained by the top-down method based on MCF inversions. With the OH constrained by precursor observations, the methane chemical loss is 471-508 Tg yr$^{-1}$ averaged from 2000 to 2009. The new adjusted estimate is more consistent with the top-down estimates in the recent global methane budget by the Global Carbon Project (GCP) (459-516 Tg yr$^{-1}$) than the bottom-up estimates using the original model-simulated OH fields (577-612 Tg yr$^{-1}$). The overestimation of global $[OH]_{trop-M}$ and methane chemical loss simulated by the atmospheric chemistry models is caused primarily by the models' underestimation of carbon monoxide and total ozone column, and overestimation of nitrogen dioxide. Our results highlight that constraining the model simulated OH fields with available OH precursor observations can help improve the bottom-up estimated methane sink.





## Introduction

Methane ($CH_4$) is a potent greenhouse gas, with its 100-year global warming potential 27 (for non-fossil $CH_4$) and 30 times (for fossil $CH_4$) times that of $CO_2$ (Forster et al., 2021). The tropospheric $CH_4$ mixing ratios have increased by more than 1.6 times between pre-industrial and the present day, resulting in 0.54 ±0.11 W m$^{-2}$ radiative forcing from 1750 to 2019 (Forster et al., 2021). After a short-term stabilization during 2000-2006, the atmospheric methane mixing ratio rose increasingly quickly from 5 ppbv yr$^{-1}$ in 2006 to 17 ppbv yr$^{-1}$ in 2021 based on surface networks (Dlugokencky, NOAA/GML,2022). The rapid growth in atmospheric $CH_4$ over the recent decade further challenges meeting the 1.5-2.0°C targets of the Paris Agreement (Nisbet et al., 2019) and therefore is becoming an increasing concern (Jackson et al. 2020).

Understanding the drivers of atmospheric methane changes rely on accurate estimates of the global methane budget, as methane concentrations in the atmosphere are the net balance between emissions and sinks. To estimate this budget, the Global Carbon Project (GCP) has established the global $CH_4$ budget by gathering up-to-date observations and model information (Kirschke et al. 2013; Saunois et al., 2016; 2017; 2020). One of the remaining largest uncertainties, as pointed out by the most recent $CH_4$ budget (Saunois et al., 2020), is the chemical loss of $CH_4$. The chemical loss of $CH_4$ stems mainly through the reaction of $CH_4$ with hydroxyl radical (OH), which is also the most important $CH_4$ sink.

The hydroxyl radical (OH) is a key species in tropospheric chemistry that reacts with most greenhouse gases and air pollutants (Levy, 1971), being the main oxidant of the lower atmosphere. Due to its extremely short lifetime (typically 1 second) and spatial variability, direct observations do not allow for the quantification of the global [OH] distributions. The [OH] for calculating the chemical sink of $CH_4$ is thus estimated either from top-down or bottom-up methods. The top-down method estimates [OH]





mainly through inversions constrained by independent observations of 1-1-1trichloroethane (methyl

chloroform, MCF), assuming that emissions of this compound are well known. Such MCF-based top-

down method has been widely used in the scientific community to derive OH trends but it can only

yield the global to latitudinal mean [OH] due to the sparse MCF observations and does not represent the

chemical feedback on OH (e.g., Prinn et al., 2001; Bousquet et al., 2005; Montzka et al., 2011; Naus et

al., 2021; Patra et al., 2020). Bottom-up approaches on the other hand simulate the [OH] by atmospheric

chemistry models to account for the chemical mechanisms that determine OH production and loss, but

their estimates of the global mean [OH] usually disagree with MCF-based estimates (Naik et al., 2013;

Zhao et al., 2019).

The global [OH] estimated by bottom-up model-based and top-down MCF-based methods are different

in magnitudes, inter-annual variations, and trends, resulting in large differences in estimated $CH_4$ sinks

between the two methods. In the last global $CH_4$ budget, most of the OH fields used to estimate the

bottom-up $CH_4$ sink were obtained from the atmospheric chemistry models that participated in the

IGAC/SPARC Chemistry-Climate Model Initiative Phase-1 (CCMI-1) project. However, these models

showed a wide range of $9.4\text{-}14.4\times10^5$ molec cm$^{-3}$ in global airmass-weighted tropospheric mean [OH]

([OH]$_{trop-M}$) (Zhao et al., 2019; Saunois et al., 2020 ), thus mostly higher than the values estimated by

the MCF-based inversions ($\sim10\times10^5$ molec cm$^{-3}$; Prinn et al., 2001; Bousquet et al., 2005). Indeed, the

mean $CH_4$ chemical loss for 2000-2009, as calculated by bottom-up approaches, is 595 Tg yr$^{-1}$ (range

489-749 Tg yr$^{-1}$), much higher than the 505 Tg yr$^{-1}$ (range 459-516 Tg yr$^{-1}$) estimated by top-down $CH_4$

inversions (Saunois et al., 2020). Those top-down inversions using box models conclude that decreased

[OH] and therefore $CH_4$ chemical loss after the mid-2000s can explain the resumed $CH_4$ increase since

2006 (Turner et al., 2017; Rigby et al., 2017) while more recent 3D inversions show no significant trend

in [OH] after 2000 (Naus et al., 2021; Patra et al., 2021). In contrast to top-down MCF-based inversions,





atmospheric chemistry models simulate a continuous increase in [OH] and consequently $CH_4$ chemical
loss from the 1980s (Zhao et al., 2020b).

Reconciling the bottom-up and top-down estimates of the methane chemical sink is essential for a more
accurate estimate of the global methane budget and to better attribute the observed changes in
atmospheric growth rates of $CH_4$. One way to improve the bottom-up estimates of the $CH_4$ sink and thus
reconcile the difference is to correct the [OH] simulated by atmospheric models using observations of
OH precursors. Indeed, uncertainties in the [OH] simulated by atmospheric models can be attributed to
biases in precursor concentrations. For example, Naik et al. (2013) found that an underestimation of
carbon monoxide (CO) in the Northern Hemisphere can contribute to the overestimation of [OH] in this
hemisphere; Strode et al. (2015) estimated that removing the model bias in $O_3$ and water vapor ($H_2O_{(g)}$)
and reducing northern hemispheric nitrogen oxides ($NO_x = NO + NO_2$) emissions can reduce a high bias in
the global OH burden by 10%. In addition, the budget analysis of OH production and loss showed that
about 90% of OH production is directly related to stratospheric and tropospheric ozone ($O_3$), $H_2O_{(g)}$, and
nitrogen oxide (NO), and ~60% of OH is removed by reaction with CO, $CH_4$ and formaldehyde ($CH_2O$)
(Lelieveld et al., 2016; Zhao et al., 2020b). Thus, bottom-up estimates of the $CH_4$ sink from chemistry
transport models can be improved if one can reduce the biases due to these OH precursors in modeled
[OH].

Fortunately, several satellites have collected long-term continuous observations of the aforementioned
OH precursors with global coverage, providing the opportunity to evaluate and improve bottom-up
estimates of the $CH_4$ sink. In this context, the main objective of this study is to reconcile the bottom-up
and top-down estimates of the $CH_4$ sink by improving the atmospheric model simulated OH fields using
multiple satellite observations and meteorological data from reanalysis. As a result, top-down estimates





of $CH_4$ emissions will also benefit from the improved 3D distributions of [OH] (Zhao et al., 2020a; Saunois et al. 2020). We first evaluate the OH precursors (CO, $CH_4$, $O_3$, $CH_2O$, and $NO_2$, the total

column $O_3$, and $H_2O_{(g)}$) simulated for the year 2010 by the CESM1-CAM4chem and GEOSCCM; these models participated in the CCMI-1 project and were used to estimate the global methane sink in Saunois et al. (2020) and represent two different chemical mechanisms. We then estimate the observation-based OH fields by correcting model biases of the two modeled OH fields due to the above-mentioned OH precursors using the Dynamically Simple Model of Atmospheric Chemical Complexity

(DSMACC). By doing so, we quantify the bias in tropospheric [OH] attributable to each precursor. Finally, we estimate the chemical sink of $CH_4$ using the observation-based OH field and, based on the uncertainties inferred for [OH], we reveal the dominant factors contributing to the uncertainties in $CH_4$ chemical sink at the global and regional scales.

## 2 Method

### 2.1 Observational data

The total column $O_3$, which mainly influences the O ($^1$D) photolysis rate, is constrained by the National Aeronautics and Space Administration (NASA) Solar Backscatter Ultraviolet (SBUV) Merged Ozone Data Set (MOD) (Frith et al., 2014). The SBUV/MOD column $O_3$ data are derived by combining

observations from nine SBUV-type instruments aboard the NASA Aura satellite. The monthly 5° zonal mean $O_3$ columns are available from January 1970 to December 2018.

Tropospheric $O_3$ is important in determining OH production. We constrain its spatial distributions using the tropospheric column $O_3$ data from the combined Aura Ozone Monitoring Instrument/Microwave

Limb Sounder (OMI/MLS) satellite observations, which are generated by subtracting the co-located MLS limb measurements (integrated over the stratosphere to derive stratospheric column ozone) from





total column ozone retrieved by OMI, an Ultraviolet/Visible nadir solar backscatter spectrometer (Ziemke et al., 2006).

The tropospheric nitrogen oxide family ($NO_x$=NO+$NO_2$) participates in both OH production (reaction of nitrogen oxide (NO) with hydroperoxyl radical ($HO_2$) or organic peroxy radicals ($RO_2$)) and loss (mainly reaction of $NO_2$ with OH). In the Dynamically Simple Model of Atmospheric Chemical Complexity (DSMACC) used in this study (see section 2.3), the $NO_x$ family is constrained by either $NO_2$ or NO concentrations. We constrain the spatial distributions of boundary layer $NO_x$ family use

satellite observations of $NO_2$ tropospheric column density use the Quality Assurance for Essential Climate Variables (QA4ECV) OMI $NO_2$ retrieval product (Boersma et al., 2018). Due to its short lifetime, $NO_x$ emitted from the surface mainly remains within the planetary boundary layer. Thus, satellite-retrieved vertical column densities are widely used in understanding the $NO_2$ distributions within the boundary layer instead of the whole troposphere (e.g., Cooper et al., 2020; Geddes et al.,

2017).

We also constrain tropospheric CO, $CH_4$, and $CH_2O$ to better represent OH loss in the troposphere. Distributions of CO and $CH_2O$ are taken from 4D variational data assimilation of tropospheric CO column retrieved from the spaceborne Measurements Of Pollution In The Troposphere instrument v7

TIR-NIR product (MOPITT v7, Deeter et al., 2017) and column $CH_2O$ OMI version product (OMI version3, González Abad et al. 2015), respectively (Zheng et al., 2019). $CH_4$ distributions are taken from data assimilation of surface $CH_4$ observations (Zhao et al., 2020a) mainly from the U.S. National Oceanic and Atmospheric Administration (NOAA/ESRL, Dlugokencky et al. (1994)). The assimilated surface CO concentration and $CH_4$ profiles show good agreement with independent ground-based

observations and aircraft observations, respectively (Zheng et al., 2019; Zhao et al., 2020a).





Meteorological conditions, mainly water vapor ($H_2O_{(g)}$) and air temperature ($T_a$) can also influence tropospheric [OH]. The $H_2O_{(g)}$ (represented as specific humidity) and $T_a$ are constrained by the second Modern-Era Retrospective analysis for Research and Applications (MERRA-2) reanalysis data from NASA's Global Modeling and Assimilation Office (GMAO) (Gelaro et al. 2017).

## 2.2 The 3D atmospheric chemistry model simulations

The 3D distributions of OH fields and OH precursors for the year 2010 are taken from the REF-C1 experiment of the IGAC/SPARC Chemistry-Climate Model Initiative Phase-1 (CCMI-1) (Hegglin and Lamarque, 2015; Morgenstern et al., 2017). The REF-C1 experiment is driven by state-of-the-art historical forcings and sea surface temperatures from observations and covers 51 years (1960-2010).

We include simulations from two models with different chemical mechanisms: (i) the Community Earth System Model (CESM) using the Community Atmosphere Model version 4 as atmosphere component (CESM1 CAM4-chem, Tilmes et al., 2015; 2016) and (ii) the GEOS-5 Chemistry Climate Model (GEOSCCM; Molod et al., 2012,2015; Oman et al., 2011, 2013). The tropospheric chemistry of CESM1 CAM4-chem is based on MOZART-4 mechanisms with minor updates (Emmons et al., 2010; Lamarque et al., 2012) and the GEOSCCM is based on the Global Modeling Initiative (GMI) chemistry and transport model (Duncan et al., 2007), which was originally developed for the GEOS-Chem model. The CO, $NO_2$, $O_3$, $CH_4$, $CH_2O$ mixing ratios, total columns $O_3$, and metrological conditions including $T_a$ and $H_2O_{(g)}$ simulated by CESM1 CAM4-chem and GEOSCCM in 2010 are compared with the observational data described in Section 2.1 in the supplementary (Fig. S1, S2, and S3). A detailed description of the two model settings related to OH and the CCMI-1 model experiments can be found in Morgenstern et al. (2017) and Zhao et al. (2019).




## 2.3 The chemical box model DSMACC

Differing from 3D atmospheric chemistry models, which simulate the [OH] by gridded emissions inventories of its precursors, a chemical box model simulates [OH] by prescribing precursor concentrations and the meteorological states. Thus, one can estimate the sensitivity of [OH] to different

precursor concentrations and meteorological parameters. Here we use the Dynamically Simple Model of Atmospheric Chemical Complexity (DSMACC; Emmerson and Evans, 2009) to estimate the sensitivity of [OH] to chemical species including $CO$, $NO_2$, $O_3$, $CH_4$, $CH_2O$, the total column $O_3$, and meteorological conditions including $T_a$ and $H_2O_{(g)}$ following the approach of Nicely et al. (2018).

The DSMACC model takes advantage of the chemical pre-processor (KPP) to generate the FORTRAN code for a chosen chemical mechanism. In this study, the DSMACC model is compiled with MOZART-4 and GEOS-Chem chemical mechanisms, respectively, to be consistent with the associated 3D models CESM1 CAM4-chem and GEOSCCM. The clear-sky photolysis rates of chemical species are estimated by the tropospheric ultraviolet and visible (TUV) radiation model. Forced by meteorological variables

($H_2O_{(g)}$, $T_a$, and pressure), total column $O_3$, and gas concentrations simulated by the CESM1 CAM4-chem and GEOSCCM, DSMACC run forward until reaching the diurnal steady state of OH. Nicely et al. (2018) have estimated the response of [OH] to changes in OH precursors by conducting DSMACC model simulations for broad latitude and pressure bins. Here we run the DAMSCC model for each model pixel of the 3D grid to better represent the heterogeneous spatial distributions of OH. For

example, the CESM1 CAM4-chem has 144 (longitude) ×96 (latitudes) ×13 (pressure level) model grids in the troposphere. For each sensitivity experiment (Sect. 2.4), we therefore conduct 179,712 DSMACC model simulations (for CESM1 CAM4-chem grid) each month.





## 2.4 DSMACC experiments

Table 1 lists the experiments conducted with the DSMACC chemical box model. The reference experiment (Ref_model in Table 1) is conducted by running DSMACC with the monthly mean chemical species concentrations and meteorological conditions simulated by the 3D models (CESM1 CAM4-chem/ GEOSCCM) for each pixel using the corresponding chemical mechanisms. In the All_obs simulation, the CO, $NO_2$, $O_3$, $CH_4$, and $CH_2O$, total column $O_3$, $T_a$, and $H_2O_{(g)}$ are replaced with the

available observational-based data (regrid to model resolution) as described in Section 2.2, while other DSMACC inputs (pressure and other chemical species) are the same as in Ref_model. The observation-based [OH] ($[OH]_{obs}$) in each 3D model pixel for two different chemical mechanisms is estimated by correcting [OH] as simulated by the corresponding 3D models ($[OH]_{model}$) by the ratio between [OH] simulated by DSMACC experiments for the All_obs ($[OH]_{DSMACC\_all\_obs}$) and for the Ref_model

($[OH]_{DSMACC\_Ref\_model}$):

$$[OH]_{obs} = [OH]_{model} \times \frac{[OH]_{DSMACC\_all\_obs}}{[OH]_{DSMACC\_Ref\_model}} \qquad (1)$$

Then, we also perform 8 sensitivity experiments (xk_obs in Table 1) that only adjust one individual

chemical species or meteorological condition (here and after represented as $x_k$) to the observations (obs), keeping the other parameters equal to the simulated values from the chemistry-climate model. Because of high computing costs, we conduct the sensitivity experiments using only CESM1 CAM4-chem outputs. The OH biases due to each factor ($\delta[OH]_k$) are estimated by introducing the [OH] simulated in the sensitivity experiment xk_obs ($[OH]_{DSMACC\_xk\_obs}$) as:


$$[OH]_{DSMACC\_xk\_obs} = [OH]_{model} \times \frac{[OH]_{DSMACC\_xk\_obs}}{[OH]_{DSMACC\_Ref\_model}} \qquad (2)$$

$$\delta[OH]_{xk} = [OH]_{model} - [OH]_{DSMACC\_xk\_obs} \qquad (3)$$





## 2.5 Chemical loss of CH$_4$

We estimate the yearly tropospheric chemical loss of CH$_4$ through reaction with OH ($L_{CH4+OH}$) at global and regional scale from 2000 to 2009 by integrating the reaction of CH$_4$ with OH:

$$L_{CH4+OH} = \sum_i \sum_t K(T)m(CH4)[OH]\delta t \tag{4}$$

Where i is the index of the model pixel and $\delta t$ is the integration time step (3 hours). The monthly 3D distributions of CH$_4$ mass (m(CH4)) during 2000-2009 are from data assimilation of surface CH$_4$

observations from NOAA/ESRL (Dlugokencky et al. 1994) and the T$_a$ distributions are from MERRA-2 reanalysis data (see Section 2.2). The reaction rate $K(Ta)$ is a function of T$_a$ as given by Sander et al. (2011):

$$K(Ta) = 2.45 \times 10^{-12} e^{-\frac{1775}{Ta}} \tag{5}$$

The contribution of each factor $x_k$ to the bias in $\delta L_{CH4+OH\_xk}$ can be estimated as:


$$\delta L_{CH4+OH\_xk} = \sum_i \sum_t K(T)m(CH4)\delta[OH]_{xk}\delta t \tag{6}$$

With $L_{CH4+OH}$, we further estimate the CH$_4$ lifetime to reaction with tropospheric OH ($\tau_{CH4}$) through the global CH$_4$ burden:

$$\tau_{CH4} = \frac{\sum_i m(CH4)}{L_{CH4+OH}} \tag{7}$$

**3 Results**

**3.1 Observation-based tropospheric [OH]**

**3.1.1 Global tropospheric OH burden.**

The global airmass-weighted tropospheric mean [OH] ([OH]$_{trop-M}$) simulated by CESM1 CAM4-chem and GEOSCCM in 2010 are $11.9\times10^5$ molec cm$^{-3}$ and $12.6\times10^5$ molec cm$^{-3}$, respectively. By adjusting

OH precursors and meteorological conditions (total column O$_3$, tropospheric O$_3$, CO, CH$_4$, CH$_2$O, boundary layer NO$_2$, H$_2$O$_{(g)}$, and T$_a$) to the observations using the DSMACC model, we estimated the





observation-based $[OH]_{trop-M}$ to be $9.9\times10^5$ molec cm$^{-3}$ and $10.4\times10^5$ molec cm$^{-3}$ with CESM1 CAM4-chem and GEOSCCM chemical mechanisms, respectively (Fig.1 and Table 2). Compared with the original OH fields simulated by CESM1 CAM4-chem and GEOSCCM, the observation-based OH fields reduce the model-simulated global $[OH]_{trop-M}$ by ~$2\times10^5$ molec cm$^{-3}$. The global $[OH]_{trop-M}$ estimated by the observation-based OH fields in this study is lower than the value estimated by Spivakovsky et al. (2000) ($11.6\times10^5$ molec cm$^{-3}$; S2000 OH field), which is used in the chemistry-transport model (CTM) intercomparison experiment (TransCom-CH4) (Patra et al., 2011) but consistent with those estimated by MCF-based inversions (~$10\times10^5$ molec cm$^{-3}$; Bousquet et al., 2005; Korl and Lelieveld, 2003). The consistency with the MCF-based estimates indicates that our approach (correcting model bias through available observations) is capable of capturing the global OH burden.

### 3.1.2 The OH spatial distribution.

Fig. 1 and Fig. 2 show the spatial distribution and zonal average of the $[OH]_{trop-M}$, respectively, estimated from the observation-based and original model simulated OH fields. The observation-based OH fields show similar spatial distributions as their respective original model simulations, with high $[OH]_{trop-M}$ ($10$-$15\times10^5$ molec cm$^{-3}$) over East Asia, South Asia, and Northern Africa, corresponding to the regions with high tropospheric $O_3$, $NO_2$, and $H_2O_{(g)}$ (Fig. S1 and Fig. S3 ). The lowest $[OH]_{trop-M}$ is found over the high latitudes ($<4\times10^5$ molec cm$^{-3}$) due to less ultraviolet radiation and over the Amazon forest ($4$-$8\times10^5$ molec cm$^{-3}$) due to high biogenic non-methane volatile organic compounds (NMVOC) emissions. The observation-based $[OH]_{trop-M}$ averaged over the northern tropics ($0$-$30°N$) and northern mid-to-high latitudes ($30$-$90°N$) are $>14\times10^5$ molec cm$^{-3}$ and $>7\times10^5$ molec cm$^{-3}$, respectively, for both chemical mechanisms, which are higher than those over the southern tropics ($12.2$-$13.6\times10^5$ molec cm$^{-3}$)



and southern mid-to-high-latitudes ($5.3-5.6 \times 10^5$ molec cm$^{-3}$, Table 3 and Fig. 2). The two observation-
based OH fields show similar mean [OH] over most of the latitudinal bands except for the southern
tropics (0-30°S), where the mean [OH]$_{trop-M}$ estimated by the GEOSCCM chemical mechanism is $1.4 \times 10^5$ molec cm$^{-3}$ higher than the one from CESM1 CAM4-chem (Table 3 and Fig. 2).

Compared to the original [OH]$_{trop-M}$ simulated by CESM1 CAM4-chem and GEOSCCM, adjusting to
observations reduces the [OH]$_{trop-M}$ by $2-8 \times 10^5$ molec cm$^{-3}$ over most regions except the tropical forests.
The reduction of mean [OH]$_{trop-M}$ over the northern tropics (0°-30°N) and mid-to-high latitudes (30°-
90°N) are $>3 \times 10^5$ molec cm$^{-3}$ and $>2 \times 10^5$ molec cm$^{-3}$, respectively, which is larger than that the over the
southern tropics (0°-30°S, by $\sim 2 \times 10^5$ molec cm$^{-3}$) and mid-to-high latitudes (30°-90°S, by $0.6 \times 10^5$
molec cm$^{-3}$). The Northern Hemisphere to Southern Hemisphere (N/S) ratios of the simulated OH fields
are reduced from 1.35 to 1.24 for CESM1 CAM4-chem, and 1.26 to 1.15 for GEOSCCM. Although the
N/S ratios of the observation-based OH fields are still higher than the 1, which is obtained from MCF-
based inversions (Bousquet et al., 2005; Patra et al., 2014), incorporating available observations has
significantly reduced the model simulated N/S ratio.

The spatial distribution of the observation-based [OH] of this study is different from the S2000 OH field.
The S2000 OH field shows a high [OH]$_{trop-M}$ over the regions with biomass burning emissions (Fig. S4).
Instead of considering the detailed spatial distributions of nitrogen oxides, Spivakovsky et al. (2000) use
a series of NO$_t$ (NO$_2$+NO+2N$_2$O$_5$+NO$_3$+HNO$_2$+HNO$_4$) profiles for land and ocean over large regions
(Fig. S5). As shown in Fig. S4 and Fig. S5, the highest [OH]$_{trop-M}$ over South America and Africa
estimated by Spivakovsky et al. (2000) correspond to high NO$_t$ mixing ratios over these two regions.
The [OH] shows high positive sensitivity to NO$_t$ in the free troposphere, due to low VOCs and NO$_t$
mixing ratios (Fig.S6). Using satellite observations, Choi et al. (2014) showed that the high NO$_2$ mixing

ratios in the free troposphere are mainly located near polluted urban regions (e.g., North America, Europe, and Asia), which is more similar to the $NO_2$ distribution simulated by 3D atmospheric models

(Fig. S1). Thus, although the S2000 OH field gives an N/S ratio of 1, its spatial distribution may have biases due to the simplification in the $NO_t$ distributions.

**3.2 Contribution from individual factors to model biases in $[OH]_{trop-M}$**

By conducting the sensitivity simulations listed in Table 1, we estimate the contribution of individual

factors to model biases based on Equation 2-3. The sensitivity of OH to model biases in tropospheric $O_3$, stratospheric $O_3$, $H_2O_{(g)}$, and $NO_x$ emissions have been tested in Strode et al. (2015) using GEOSCCM. In this Section, we extend the procedure of Strode et al. (2015) by including more factors: $T_a$, CO, $CH_2O$, $CH_4$, and $NO_2$ in the boundary layer.

Table 4 summarizes the contribution of each chemical precursor and meteorological condition to the difference between CESM1 CAM4-chem simulated and observation-based global $[OH]_{trop-M}$. The total contribution of the 8 individual factors to the difference in global $[OH]_{trop-M}$ estimated from the simulation xk_obs is $2.0 \times 10^5$ molec cm$^{-3}$ (Table 4), consistent with that estimated from the simulation All_obs (Table 2), indicating that the nonlinear effect of atmospheric chemistry is negligible on the

global scale. Indeed, although the atmospheric OH is produced and removed through complex nonlinear chemical reactions, one can infer the large-scale $[OH]_{trop-M}$ changes by roughly summing the influence from individual factors.

**3.2.1 Contribution from CO**

CO is the largest OH sink in the troposphere (Lelieveld et al., 2016; Zhao et al., 2020b). The sensitivity simulation CO_obs shows that a 1 ppbv increase in CO can result in a decrease in [OH] by up to more



than $3\times10^4$ molec cm$^{-3}$ (Fig. S6). Compared with the CO distributions from inversions that assimilated MOPITT observations, the CESM1 CAM4-chem underestimates the global tropospheric mean CO mixing ratio by 24 ppbv (Fig. S1). Based on the DSMACC simulations CO_obs and Ref_CESM (Table 1), we find that the negative bias in CO contributes most to the difference in the modeled versus observation-based [OH]$_{trop-M}$ ($1.3\times10^5$ molec cm$^{-3}$; Table 4 and Figure 3). The underestimation of CO is common in atmospheric models and was treated either as a cause or an effect of the overestimated [OH] in previous studies (Naik et al., 2013; Monks et al. 2015; Stode et al., 2015). For example, based on the ACCMIP simulations, Naik et al. (2013) found that the positive bias in [OH] was most likely due to the underestimation of CO when compared with both satellite and surface observations. In contrast, Strode et al. (2015) did sensitivity simulations using the GEOSCCM model and showed that reducing OH bias could improve the accuracy of modeled CO. In this study, we do not intend to solve the cause/effect issue between CO and OH, since the discrepancy in [OH]$_{trop-M}$ of $1.3\times10^5$ molec cm$^{-3}$ could also be understood as the global tropospheric [OH] changes that would be needed to simulate the observed CO.

The underestimation of the CO mixing ratio is larger over the Northern Hemisphere (30 ppbv) than over the Southern Hemisphere (18 ppbv) (Fig.S1). The largest bias in [OH]$_{trop-M}$ induced by CO is found over the northern tropics ($1.9\times10^5$ molec cm$^{-3}$) followed by those over the northern mid-to-high latitude regions and the southern tropics ($1.2\times10^5$ molec cm$^{-3}$; Table 4 and Fig. 2). Naik et al. (2013) demonstrated that the model bias in CO contributes to the overestimation of the modeled N/S ratio in [OH]$_{trop-M}$. In this study, although the underestimation of CO leads to a larger positive bias of [OH]$_{trop-M}$ over the Northern Hemisphere than the Southern Hemisphere, the observation-based adjustment only reduces the positive bias of the N/S ratio by 0.02. This means that the N/S difference of the CO bias is not sufficient to explain the inconsistency between the CESM1 CAM4-chem simulated and MCF-based N/S ratio in [OH]$_{trop-M}$.



### 3.2.2 Contribution from tropospheric $O_3$

Tropospheric $O_3$ can contribute to both primary and secondary OH production. Compared to satellite observations from OMI, CESM1 CAM4-chem simulations show a large overestimation of tropospheric $O_3$ over the 15-60°N (up to 14 DU, 40%) and an underestimation (14 DU, 40%) over the tropics and Southern Hemisphere (Fig. S1).

At the global scale, the model bias in $O_3$ leads to a negative bias on $[OH]_{trop-M}$ by $0.3\times10^5$ molec cm$^{-3}$, much smaller than that caused by CO (Table 4). However, at the regional scale, The CESM1 CAM4-chem simulated $[OH]_{trop-M}$ is enhanced by ~$1\times10^5$ molec cm$^{-3}$ over the tropics (15°S-15°N) and ~$0.5\times10^5$ molec cm$^{-3}$ over the mid-southern hemisphere (15°-60°S), while it is reduced by $0.1-0.3\times10^5$ molec cm$^{-3}$ over the mid-northern hemisphere (15°-60°N) when adjusted to OMI/MLS tropospheric column $O_3$ (Fig. 2). The adjustment reduces the N/S ratio of $[OH]_{trop-M}$ by 0.07, still cannot explain the overestimation of the N/S ratio but leads to a larger correction than the one with CO alone.

### 3.2.3 Contribution from boundary layer $NO_2$.

The sensitivity of OH to $NO_2$ is highly variable. We estimate that a 1 ppbv increase in $NO_2$ can lead to a change of [OH] ranging from $-3\times10^6$ molec cm$^{-3}$ to more than $+10\times10^6$ molec cm$^{-3}$, depending on the mixing ratio of NMVOCs (represented as HCHO+isoprene) and $NO_2$ (Fig. S6). Compared to the QA4ECV $NO_2$ retrieval product, CESM CAM-Chem overestimates tropospheric $NO_2$ over most regions except North China and tropical forests (Fig. S1).



At the global scale, the overestimation of $NO_2$ leads to a positive bias in $[OH]_{trop-M}$ by $0.3\times10^5$ molec cm$^{-3}$ (Table 4). At the regional scale, correcting for the PBL $NO_2$ does not influence the N/S ratio of OH. As shown in Fig. 3, the overestimation of PBL $NO_2$ results in positive bias in $[OH]_{trop-M}$ over most of the continental regions. Over tropical and temperate oceans, one can also see that the slight overestimation in $NO_2$ leads to a significate positive bias in OH by $0.5-1\times10^5$ molec cm$^{-3}$ since the sensitivity of $[OH]_{trop-M}$ to $NO_2$ can be very high ($10^7$ molec cm$^{-3}$/ ppbv $NO_2$) over the regions with low $NO_x$ and NMVOC mixing ratios. Over North China, although the model shows a large underestimation in $NO_2$ (Fig.S1), the $[OH]_{trop-M}$ is slightly smaller after adjustment. This is because over high $NO_2$ regions, the [OH] is not sensitive to an increase in $NO_2$ or even shows a negative response (Fig. S6).

### 3.2.4 Contribution from total column $O_3$

The total column $O_3$ mainly influences $O^1(D)$ photolysis through absorbing UV radiation. The CESM1 CAM4-chem mainly underestimates the total $O_3$ columns by up to ~10 DU over tropical regions compared with the SBUV/MOD observations (Fig. S2). On a global scale, the underestimation of the $O_3$ total column can lead to an overestimation of the $[OH]_{trop-M}$ by ~$0.6\times10^5$ molec cm$^{-3}$ (Fig. 2 and Fig. 3), comparable with that due to tropospheric $O_3$.

### 3.2.5 Contribution from $CH_4$ and $CH_2O$

In CCMI-1 simulations, atmospheric chemistry models prescribe the lower boundary conditions for $CH_4$ following the Representative Concentration Pathway (RCP6.0). Compared to the posterior $CH_4$ fields from inversions by assimilating the surface $CH_4$ observations, the tropospheric mean $CH_4$ mixing ratios used in the CESM CAM4-chem are ~80 ppbv lower over the tropical and extratropical regions with high biomass burning and anthropogenic emissions, and ~40 ppbv lower over other regions. But due to





the low sensitivity of [OH] to $CH_4$ changes (Fig. S1), the underestimation in $CH_4$ only leads to a small positive bias in the global tropospheric mean [OH] by $0.1 \times 10^5$ molec $cm^{-3}$.

CESM CAM4-chem overestimates $CH_2O$ by more than 50% over land but slightly underestimates $CH_2O$ over tropical oceans (Fig.S1). Since the $CH_2O$ contributes to only a small part (6%) of the total [OH] loss (Zhao et al., 2020b), such a large bias in the CESM CAM4-Chem simulated $CH_2O$ only leads to a small positive bias global mean $[OH]_{trop-M}$ by $0.1 \times 10^5$ molec $cm^{-3}$ (Fig.3).

### 3.2.6 Contribution from meteorological conditions

$H_2O_{(g)}$ is a major OH precursor that contributes to the primary production of OH and $T_a$ can influence OH production and loss rates. Compared to MERRA2 reanalysis data, CESM CAM4-chem overestimates zonally averaged $H_2O_{(g)}$ mixing ratios near the surface and around 800 hPa by ~1.5g/kg (Fig. S3). The sensitivity experiments show that a change in specific humidity by 1g/kg can lead to a change in [OH] by $>3 \times 10^5$ molec $cm^{-3}$ over the regions with high $O(^1D)$ photolysis and low NMVOC

mixing ratios (Fig. S6). As shown in Table 4 and Fig.3, globally, the model bias in $H_2O_{(g)}$ only leads to a small bias ($0.1 \times 10^5$ molec $cm^{-3}$) in $[OH]_{trop-M}$, but regionally, the model bias in $H_2O_{(g)}$ can lead to a bias in $[OH]_{trop-M}$ by the magnitude of $5.0 \times 10^5$ molec $cm^{-3}$, even larger than that induced by the bias in CO. For $T_a$, the model only shows a small bias (<1℃) compared with MERRA2 reanalysis data (Fig. S3). Thus, model bias in $[OH]_{trop-M}$ induced by $T_a$ is negligible (Fig. 4).

### 3.3 Chemical sinks of $CH_4$ as estimated by observation-based OH fields
### 3.3.1 Global and regional OH chemical sink of $CH_4$





Using the observation-based OH field, we estimate that the global tropospheric $CH_4$ loss by reaction with tropospheric OH ($L_{CH4+OH}$) averaged during the period 2000 through 2009 is 434 Tg yr$^{-1}$ and 461

440        Tg yr$^{-1}$ for CESM1 CAM4-chem and GEOSCCM, respectively. These estimates are about 105 Tg yr$^{-1}$ lower than estimated by the original model simulated OH fields (540 Tg yr$^{-1}$ and 565 Tg yr$^{-1}$, respectively; Table 2). The corresponding $CH_4$ lifetimes against tropospheric OH loss estimated by the two observation-based OH fields are 11.4 yr and 10.7 yr for CESM1 CAM4-chem and GEOSCCM, respectively, well within the range estimated by Prather et al. (2012) based on the MCF-inversions (11.2

$\pm 1.3$yr) and much longer than estimated by the original model simulated OH fields (9.1 yr for CESM1 CAM4-chem and 8.7 yr for GEOSCCM).

As shown in Table 3, more than 70% of the tropospheric $L_{CH4+OH}$ occurs over tropical regions mainly due to both high [OH] and $T_a$. Constraining the tropospheric [OH] by precursor concentrations reduces

the tropospheric $L_{CH4+OH}$ by ~30 Tg yr$^{-1}$ (16%) over the southern tropics, ~50 Tg yr$^{-1}$(21%) over the northern tropics, and ~25 Tg yr$^{-1}$(25%) over the northern mid-to-high latitude as estimated by both CESM1 CAM4-chem and GEOSCCM OH fields. Over the southern mid-to-high latitude regions, there are only a few changes (6 Tg yr$^{-1}$) in tropospheric $L_{CH4+OH}$. Thus, constraining tropospheric [OH] by precursor concentrations changes the inter-hemispheric distribution of $L_{CH4+OH}$. The values of $L_{CH4+OH}$

estimated by the observation-based OH fields are ~35 Tg yr$^{-1}$ and ~75 Tg yr$^{-1}$ lower than that estimated by the corresponding original model simulated OH fields over the Southern and Northern Hemispheres, respectively (Table 3). Thus, the inter-hemispheric difference of $L_{CH4+OH}$ (north − south) estimated by observation-based OH fields (60 Tg yr$^{-1}$ by CESM1 CAM4-chem and 48 Tg yr$^{-1}$ by GEOSCCM) is ~40% lower than estimated by the original model simulated OH fields (98 Tg yr$^{-1}$ by CESM1 CAM4-chem

and 81 Tg yr$^{-1}$ by GEOSCCM).



### 3.3.2 Global total chemical sink of CH$_4$

We estimate the global total CH$_4$ chemical sink for 2000-2009 by gathering: (1) the tropospheric L$_{CH4+OH}$ estimated using the original model-simulated and observation-based OH fields, (2) the CH$_4$ loss

in the stratosphere (26 Tg yr$^{-1}$ estimated by CESM1 CAM4-chem and 36 Tg yr$^{-1}$ estimated by GEOSCCM simulations) and CH$_4$ oxidated by chlorine (11 Tg yr$^{-1}$) given by Saunois et al. (2020). We then compare the chemical sink estimated in this study with that estimated by the bottom-up and top-down methods given by previous GCP global CH$_4$ budget (Saunois et al., 2016; 2020).

As shown in Fig. 4, the bottom-up estimates in the GCP global CH$_4$ budget (blue bars) have a large range (483-738 Tg yr$^{-1}$ in Saunois et al. (2016) and 489-779 Tg yr$^{-1}$ in Saunois et al. (2020)), much higher than those from the top-down method (514 Tg yr$^{-1}$ in Saunois et al. (2016) and 459-516 Tg yr$^{-1}$ in Saunois et al. (2020)). The CH$_4$ sinks simulated by CESM1 CAM4-chem (549 Tg yr$^{-1}$) and GEOSCCM (585 Tg yr$^{-1}$) were included in the bottom-up estimates in Saunois et al. (2020) (green bar) and is

slightly lower than the average value estimated using different OH fields (595 Tg yr$^{-1}$).

In this study, the global total CH$_4$ chemical sinks estimated using the originally simulated tropospheric OH and constrained CH$_4$ mixing ratios are 577 Tg yr$^{-1}$ and 612 Tg yr$^{-1}$ for CESM1 CAM4-chem and GEOSCCM, respectively, close to the mean values estimated by the bottom-up method (around 600 Tg

480 yr$^{-1}$) using different OH fields but much higher than the top-down estimates (around 500 Tg yr$^{-1}$). It should be noted that the bottom-up estimates of the chemical loss of CH$_4$ in previous GCP global CH$_4$ budget were calculated using model-simulated CH$_4$ mixing ratios (Fig.4; Saunois et al. 2020). The CH$_4$ mixing ratios simulated by CESM1 CAM4-chem and GEOSCCM are lower than that used in this study (Fig. S1). Thus, the chemical sink of CH$_4$ estimated in this study is higher than that estimated in Saunois

et al. (2020) by ~30 Tg yr$^{-1}$. After adjusting the main OH precursors to observations, the global





chemical sink of $CH_4$ is 471-508 Tg yr$^{-1}$, as estimated using the two observation-based OH fields, more consistent with top-down method estimates (~500Tg yr$^{-1}$).

The above analyses show that the large uncertainties in the bottom-up estimates of the $CH_4$ chemical sink are attributable to the use of the model-simulated OH fields with known biases. Constraining the OH field with available precursor observations to correct the global [OH], the magnitude of the methane loss is more in line with top-down methane inversions. Therefore, we partly reconcile the bottom-up and top-down estimates of the $CH_4$ sink. Although only two of seven bottom-up models synthesized in Saunois et al. (2020) are considered in this study, our approach can be generalized to other chemistry-climate models. Instead of directly using the OH fields simulated from an atmospheric chemistry model, the bottom-up estimates can use the precursor observations and box-model based approach proposed here to reduce model biases of OH fields.

### 3.3.3 Contribution from the model biases of individual OH precursors to chemical sink of $CH_4$

We further quantify the influence of model biases in individual OH precursors on the bottom-up estimates of $CH_4$ chemical sink ($\delta L_{CH4+OH\_xk}$). At the global scale, the underestimation of CO and total column $O_3$ and the overestimation of $NO_2$ by the CESM1 CAM4-chem lead to a positive bias of 60 Tg yr$^{-1}$(11%), 22 Tg yr$^{-1}$(4%), and 22 Tg yr$^{-1}$(4%) in tropospheric $L_{CH4+OH}$ (Fig.4 and Table 4), respectively, while an underestimation of tropospheric $O_3$ leads to a negative bias of 17 Tg yr$^{-1}$ (3%) in tropospheric $L_{CH4+OH}$. Although the model bias of $[OH]_{trop-M}$ induced by $H_2O_{(g)}$ is negligible on the global scale, the observation-based adjustment of $H_2O_{(g)}$ leads to a reduction in tropospheric $L_{CH4+OH}$ by 10 Tg (2%), since the model overestimation of $H_2O_{(g)}$ is concentrated over the mid-to-low latitude regions where tropospheric $CH_4$ oxidation mainly occurs (Fig. S3). The model bias in $CH_2O$ and $CH_4$ itself leads to a small positive bias of ~1% respectively on $L_{CH4+OH}$.





As the tropospheric $L_{CH4+OH}$ mainly occurs over the mid-to-low latitude regions, the biases in $[OH]_{trop-M}$ over the high latitudes (north of 60°N or south of 60°S) due to an overestimation of CO and underestimation of $H_2O_{(g)}$ (Fig. 2), do not substantively contribute to the bias in $L_{CH4+OH}$ (Fig. 5). Over the regions north of 15°N, nearly all the precursors considered in this study contribute to the

515 overestimation of $L_{CH4+OH}$ (55 Tg yr$^{-1}$ in total), of which 47% (26 Tg yr$^{-1}$) is contributed by model underestimation of CO. South of 15°N, the underestimation of tropospheric $O_3$ results in an underestimation of $L_{CH4+OH}$ by 22 Tg yr$^{-1}$, partly offsetting the overestimation of $L_{CH4+OH}$ induced by CO (34 Tg yr$^{-1}$) and other precursors (40 Tg yr$^{-1}$ in total) (Fig. 5). As aforementioned, the inter-hemispheric difference of $L_{CH4+OH}$ derived from the observation-based OH fields is 48 Tg yr$^{-1}$ smaller than estimated

using the OH field originally simulated by CESM1 CAM4-chem. The biases in CO, tropospheric $O_3$, and boundary layer $NO_2$, lead to an overestimation of the inter-hemispheric difference of tropospheric $L_{CH4+OH}$ by 15 Tg yr$^{-1}$, 15 Tg yr$^{-1}$, and 9 Tg yr$^{-1}$, respectively, dominant the bias in the inter-hemispheric difference in tropospheric $L_{CH4+OH}$,

**4 Conclusion and discussion**

In this study, we aim to reconcile the top-down and bottom-up estimates of the global $CH_4$ sink and to quantify the contribution of each factor to the overestimation of tropospheric [OH] that is generally found in atmospheric chemistry models and to the consequent overestimation of $CH_4$ chemical loss in the bottom-up method. To do so, we propose a new approach based on precursor observations and a

530 chemical box model, to improve the 3D distributions of tropospheric OH radicals issued from atmospheric chemistry models.





We estimate two 3D observation-based OH fields based on three components: (i) simulated tropospheric [OH] and related chemical species from global 3D atmospheric chemistry models (here CESM1-CAM4chem and GEOSCCM), (ii) the sensitivities of tropospheric [OH] to its precursors in each model grid cell estimated by the chemical box model DSMACC using a chemical mechanism similar to the 3D model, and (iii) observations of chemical species related to OH production and loss (CO, $O_3$, boundary layer $NO_2$, $CH_4$, $CH_2O$, and total column $O_3$) and meteorological conditions ($H_2O_{(g)}$ and $T_a$). The chemical box model DSMACC can be compiled using different chemical mechanisms, making it possible to apply this approach to other atmospheric chemistry models and improve the OH.

The global $[OH]_{trop-M}$ estimated from observation-based OH fields is $\sim 10 \times 10^5$ moelc $cm^{-3}$ in 2010 based on two different chemical mechanisms, which is $2 \times 10^5$ molec $cm^{-3}$ lower than the original model-simulated global $[OH]_{trop-M}$, consequently reaching consistency with the value derived by MCF-based inversions (around $10 \times 10^5$ molec $cm^{-3}$; Bousquet et al., 2005; Korl and Lelieveld, 2003). The observation-based adjustments also change the latitudinal distribution of [OH], reducing the it's north to south ratios from 1.35 and 1.26 to 1.24 and 1.15 for CESM1 CAM4-chem and GEOSCCM, respectively, closer to the one obtained from MCF-based inversions (slightly smaller than 1).

Based on the simulations from CESM1 CAM4-chem, globally, the overestimation of $[OH]_{trop-M}$ arises mainly from the underestimation of CO and total column $O_3$, and the overestimation of boundary layer $NO_2$, which contribute $1.3 \times 10^5$ molec $cm^{-3}$, $0.4 \times 10^5$ molec $cm^{-3}$, and $0.3 \times 10^5$ molec $cm^{-3}$, respectively, to the bias in $[OH]_{trop-M}$. For the N/S ratio of $[OH]_{trop-M}$, the positive bias in $[OH]_{trop-M}$ over the Northern Hemisphere ($0.1-0.3 \times 10^5$ molec $cm^{-3}$) and the negative bias over the tropics and Southern Hemisphere ($0.5-1.0 \times 10^5$ molec $cm^{-3}$) due to tropospheric $O_3$ dominate the higher N/S ratio of $[OH]_{trop-M}$ estimated by the CESM1 CAM4-chem than the observation-based OH field. At the regional scale, the model bias


in $H_2O_{(g)}$ can lead to bias in $[OH]_{trop-M}$ even larger than that induced by CO.

The global $CH_4$ loss by reaction with tropospheric OH ($L_{CH4+OH}$) estimated from the observation-based OH fields is 434 Tg yr$^{-1}$ and 461 Tg yr$^{-1}$ for CESM1 CAM4-chem GEOSCCM, respectively, averaged over 2000 to 2009, which is lower than that estimated from the original model simulated OH fields by around 105 Tg yr$^{-1}$. Based on the results from CESM1 CAM4-chem, at the global scale, the underestimation of CO and total column $O_3$, and overestimation of $NO_2$ lead to positive biases in tropospheric $L_{CH4+OH}$ by 60 Tg yr$^{-1}$(11%), 22 Tg yr$^{-1}$(4%), and 22 Tg yr$^{-1}$(4%), respectively, while an underestimation of tropospheric $O_3$ leads to a negative bias in tropospheric $L_{CH4+OH}$ by 17 Tg yr$^{-1}$(3%). The inter-hemispheric difference in the tropospheric $L_{CH4+OH}$ is therefore reduced by 40% (around 35 Tg yr$^{-1}$) when estimated using the observation-based OH field. Although the bias in the N/S ratio of $[OH]_{trop-M}$ is dominated by the tropospheric $O_3$ concentration, the positive bias in the inter-hemispheric difference of $L_{CH4+OH}$ is determined together by the biases in CO (15 Tg yr$^{-1}$), tropospheric $O_3$ (15 Tg yr$^{-1}$), and boundary layer $NO_2$ (9 Tg yr$^{-1}$).

Using the tropospheric $L_{CH4+OH}$ estimated with our observation-based OH fields, the global total $CH_4$ chemical sink is 471-508 Tg yr$^{-1}$. This quantification is more consistent with top-down estimates in the previous GCP global $CH_4$ budget (459-516 Tg yr$^{-1}$, Saunois et al., 2016; 2020) than it was before the adjustment (577-612 Tg yr$^{-1}$). The bottom-up method in the previous GCP global $CH_4$ budget estimated the $CH_4$ chemical sink directly using the OH fields simulated by atmospheric chemistry models. However, the uncertainties in the model simulated OH lead to an unreliable range in the bottom-up estimated $CH_4$ chemical sink, much higher than that estimated by the top-down method. Our results highlight that constraining the OH fields using available precursor observations can improve the bottom-up estimates of the $CH_4$ sink and help reconcile the difference between the top-down and





bottom-up estimates of the CH$_4$ sink.

Although the observation-based 3D OH fields presented in this study can capture the global tropospheric OH burden and chemical loss of CH$_4$, unresolved uncertainties remain. For example, the

difference in global [OH]$_{\text{trop-M}}$ between the two observation-based OH fields estimated from CESM1 CAM4-chem and GEOSCCM simulations is $0.6 \times 10^5$ molec cm$^{-3}$. Such a difference is partly be attributable to their differences in chemical mechanisms. As discussed in Murray et al. (2021), the inter-model differences in tropospheric [OH] and its responses to precursors are largely determined by the oxidative efficiency of NMVOCs and the lifetime of NO$_x$ simulated by the models. Reducing the

uncertainties due to the modeled chemical mechanisms relies on additional observations to improve the simulation of NMVOCs oxidative efficiency and NO$_x$ lifetime.

In addition, the data that we use to constrain the OH precursors come mainly from satellite observations and reanalysis data, of which the uncertainties are not considered in this study. For example, the

MERRA-2 reanalysis data significantly overestimate H$_2$O$_{(g)}$ in the upper troposphere (Jiang et al., 2015); The QA4ECV tropospheric NO$_2$ vertical column density is lower compared with surface observations under the extreme high-pollution case compared with surface observations (Compernolle et al., 2020). Therefore, the performance of this method thus depends on the accuracy of observations used to constrain individual factors. Since the sensitivity of [OH] to each precursor is estimated by the chemical

box model, we can easily improve the [OH] using the updated observational datasets.

Another key factor that could influence the tropospheric OH burden but is unconstrained in this study is NO$_2$ in the free troposphere. Although the NO$_2$ mixing ratio is usually less than 1 ppbv in the free troposphere, the sensitivity of [OH] to NO$_2$ can be very high in low NO$_2$ regions. However, a potential



model bias due to lightning $NO_x$ emissions, which had proven to contribute significantly to the upper-tropospheric [OH] burden (Murray et al., 2013; Turner et al., 2018), is not adjusted in our study. Satellite retrievals for upper tropospheric $NO_2$ (e.g. Belmonte Rivas et al., 2015; Marais et al., 2021) could help quantify [OH] biases due to free tropospheric $NO_2$ and the contribution of lightning $NO_x$ emissions.


The observation-based adjustment reduces the simulated N/S ratio of [OH]$_{trop-M}$ by 0.1 only, which is still higher than the one obtained from MCF-inversions (less than 1.0). This difference indicates that the overestimation of N/S ratio by atmospheric models cannot be fully explained by the underestimation of CO and overestimation of $O_3$ over the Northern Hemisphere as mentioned in previous studies (Naik et

al. 2013). Including the chemical mechanism such as OH recycling by isoprene (Lelieveld et al. 2008) may help further reduce the N/S ratio for model-simulated OH fields.

The new approach proposed here to improve the 3D OH fields and chemical loss of $CH_4$ can be applicable broadly. It relies on observations of OH precursor concentrations that can be applied

efficiently to any atmospheric chemistry model with a box-model (0D) available. Here we only apply this method to two models for one year (2010) and both of them agree with MCF-based inversions in terms of the global OH burden. One future research development is to generate observation-based OH fields for all the atmospheric chemistry models included in the GCP global $CH_4$ budget and over a longer time period to see if higher consistency can also be achieved on longer timescales. It will also be

important to assess how much uncertainty in [OH] means and trends can be further reduced and achieved in detail.

Also, the $CH_4$ emissions from top-down approaches used mostly a single OH field from Spivakosky

(2000), which is climatological data without any interannual variations. Some $CH_4$-inversions used the

OH fields from chemistry-climate or chemistry-transport models with the known aforementioned biases

that may lead to bias in the inverted surface $CH_4$ fluxes. Our OH product could be used instead in $CH_4$

inversions to better infer $CH_4$ emissions and reduce the uncertainties in the global methane budget.

Further work is necessary to consider the interannual changes in our observation-based estimates.

**Data availability**

The GEOSCCM OH fields are available at the Centre for Environmental Data Analysis

(CEDA; http://data.ceda.ac.uk/badc/wcrp-ccmi/data/CCMI-1/output; last access: December 2019,

Hegglin and Lamarque, 2015), the Natural Environment Research Council's Data Repository for

Atmospheric Science and Earth Observation. The CESM1 CAM4-chem outputs for CCMI are available

at http://www.earthsystemgrid.org (Climate Data Gateway at NCAR, last access: December 2019).

**Code availability**

The DSMACC model code and descriptions are available at http://wiki.seas.harvard.edu/geos-
chem/index.php/Main_Page.


**Author contributions**

YZ, MS, and PB designed the study, analyzed data, and wrote the manuscript. XL helped with data

preparation. JGC and RBJ provided input into the study design and discussed the results. MIH provided

CCMI model outputs. BZ provided the assimilated CO and $CH_2O$ distributions. All co-authors

commented on the manuscript.

**Acknowledgements**



This work benefited from and is a contribution to the Global Methane Budget activity of the Global Carbon Project.

We acknowledge the modeling groups for making their simulations available for this analysis, the joint WCRP SPARC/IGAC Chemistry–Climate Model Initiative (CCMI) for organizing and coordinating the model simulations and data analysis activity, and the British Atmospheric Data Centre (BADC) for collecting and archiving the CCMI model output. JGC acknowledges support from the National Environmental Science Program – Climate Systems Hub. RBJ acknowledges the U.N. Environment Programme for support of the Global Methane Budget.

**Competing interests**

The authors declare that they have no conflicts of interest.

**Financial support**

This research has been supported by Shandong Provincial Natural Science Foundation (grant no. 2022HWYQ-066) and the Gordon and Betty Moore Foundation (grant no. GBMF5439), "Advancing Understanding of the Global Methane Cycle", and the U.N. Environment Programme.

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





**Table 1.** The DSMACC model experiments.

| Species | Simulations | Description |
|---|---|---|
| Ref | Ref_model | Chemical species and meteorological conditions from 3D model simulations for 2010. |
| All | All_obs | All the chemical species and meteorological conditions listed below adjusted to match observations. |
| $NO_2$ | NO2_obs_PBL | Adjust boundary layer $NO_2$ to match the OMI QA4ECV product. |
| $O_3$ | O3_obs | Adjust tropospheric $O_3$ to match OMI/MLS product. |
| $CH_4$ | CH4_obs | Adjust tropospheric $CH_4$ to match the assimilated data. |
| CO | CO_obs | Adjust tropospheric CO to match the assimilated data. |
| $CH_2O$ | CH2O_obs | Adjust tropospheric $CH_2O$ to match the assimilated data. |
| $O_3$ column | O3col_obs | Adjust total ozone column to match SBUV/MOD data. |
| $H_2O_{(g)}$ | H2O_obs | Adjust water vapor to MERRA-2 data. |
| Ta | Ta_obs | Adjust air temperatures to MERRA-2 data. |

**Table 2.** Modeled and observation-based estimates of global $[OH]_{trop-M}$, $CH_4$ sink by tropospheric OH ($L_{CH4+OH}$) averaged during 2000-2009, and the $CH_4$ lifetime against tropospheric OH.

| | | $[OH]_{trop-M}$ ($10^5$molec cm$^{-3}$) | $L_{CH4+OH}$ (Tg yr$^{-1}$) | $CH_4$ lifetime(yr) |
|---|---|---|---|---|
| Modeled | CESM1-CAM4chem | 11.9 | 540 | 9.1 |
| | GEOSCCM | 12.6 | 565 | 8.7 |
| Observation-based | CESM1-CAM4chem | 9.9 | 434 | 11.4 |
| | GEOSCCM | 10.4 | 461 | 10.7 |

**Table 3.** The modeled and observation-based $[OH]_{trop-M}$ (in $10^5$molec cm$^{-3}$) averaged over latitudinal bands during 2000-2009. The corresponding tropospheric $CH_4$ sink by OH ($L_{CH4+OH}$) (in Tg yr$^{-1}$) is given in brackets.

| | | 90°-30°S | 30°S-0° | 0°-30°N | 30°-90°N |
|---|---|---|---|---|---|
| Modeled | CESM1-CAM4chem | 5.9(49) | 14.2(173) | 17.8(226) | 9.4(93) |
| | GEOSCCM | 6.2(50) | 16.1(192) | 18.5(229) | 9.6(94) |
| Observation-based | CESM1-CAM4chem | 5.3(42) | 12.2(144) | 14.5(178) | 7.2(69) |
| | GEOSCCM | 5.6(46) | 13.6(161) | 14.9(183) | 7.4(72) |





**Table 4.** Contributions from individual factors to the difference in global $[OH]_{trop-M}$ and tropospheric $CH_4$ sink by reaction with OH between CESM1-CAM4chem simulated and the corresponding observation-based OH fields (modeled－observation-based).

|  | $[OH]_{trop-M}$ ($10^5$ molec cm$^{-3}$) | $CH_4$ sink ($Tg\ yr^{-1}$) |
|---|---|---|
| **$H_2O_{(g)}$** | 0.1 | 10 |
| **$T_a$** | 0 | 0 |
| **Column $O_3$** | 0.4 | 22 |
| **CO** | 1.3 | 60 |
| **$O_3$** | -0.3 | -17 |
| **$NO_2$** | 0.3 | 22 |
| **$CH_4$** | 0.1 | 5 |
| **$CH_2O$** | 0.1 | 6 |
| **Total** | 2.0 | 108 |

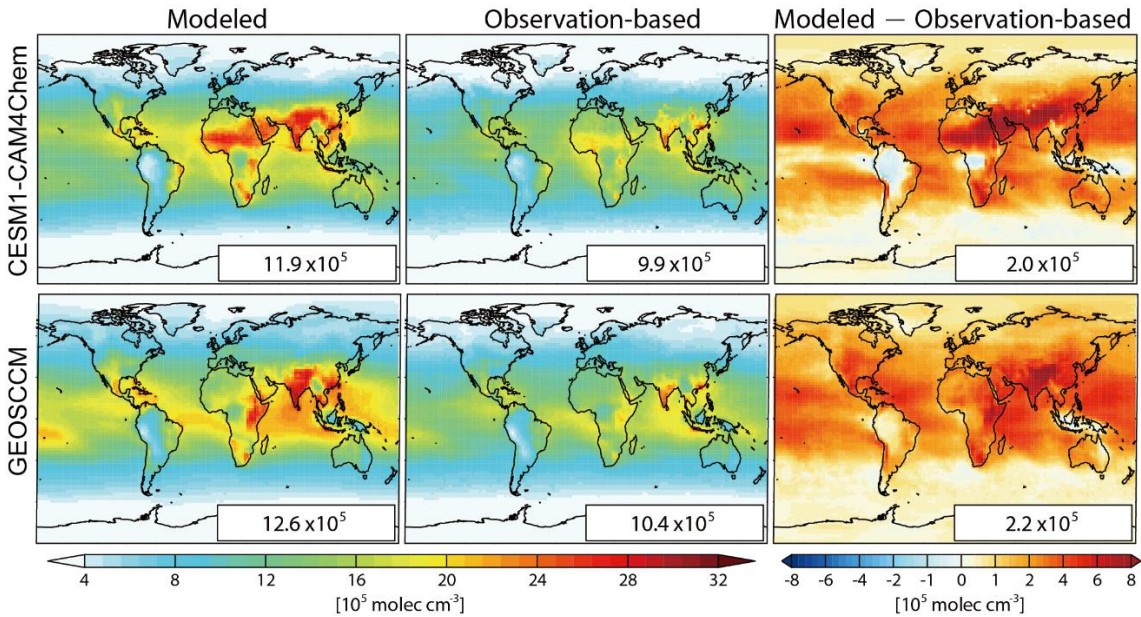

**Figure 1.** Spatial distributions of air mass-weighted tropospheric mean [OH] ($[OH]_{trop-M}$) in 2010 from model simulations (left) and constrained by observations (middle), and the difference between modeled and observation-based $[OH]_{trop-M}$ (right) estimated from CESM1-CAM4Chem (top) and GEOSCCM (bottom) simulations. The global mean values are shown inset in molec cm$^{-3}$.





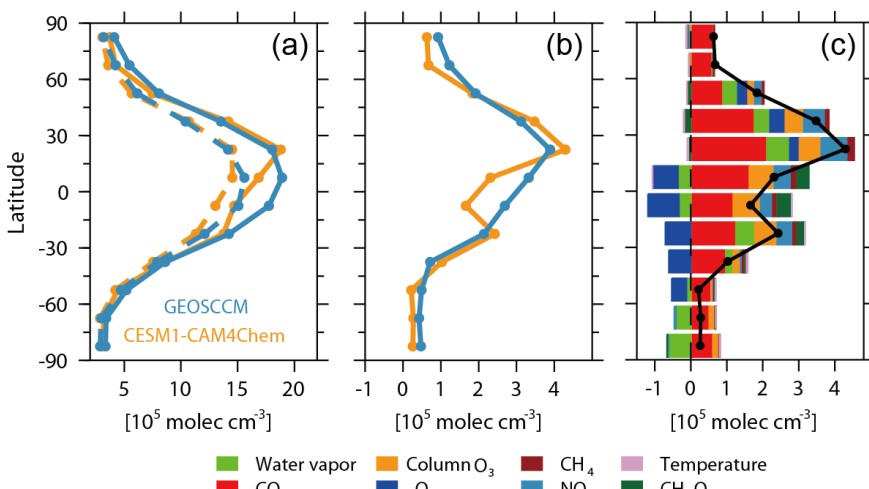

**Figure 2.** (a) Zonal averaged $[OH]_{trop-M}$ of modeled (solid lines) and observation-based OH field (dashes lines) estimated from CESM1 CAM4-chem (yellow) and GEOSCCM (blue) simulations. (b) Difference of zonal averaged $[OH]_{trop-M}$ between modeled and observation-based OH fields. (c) Difference between CESM1 CAM4-chem simulated and observational-based zonal averaged $[OH]_{trop-M}$ (black line) and the contribution from each OH precursor (colored bars) for zonal averaged difference.

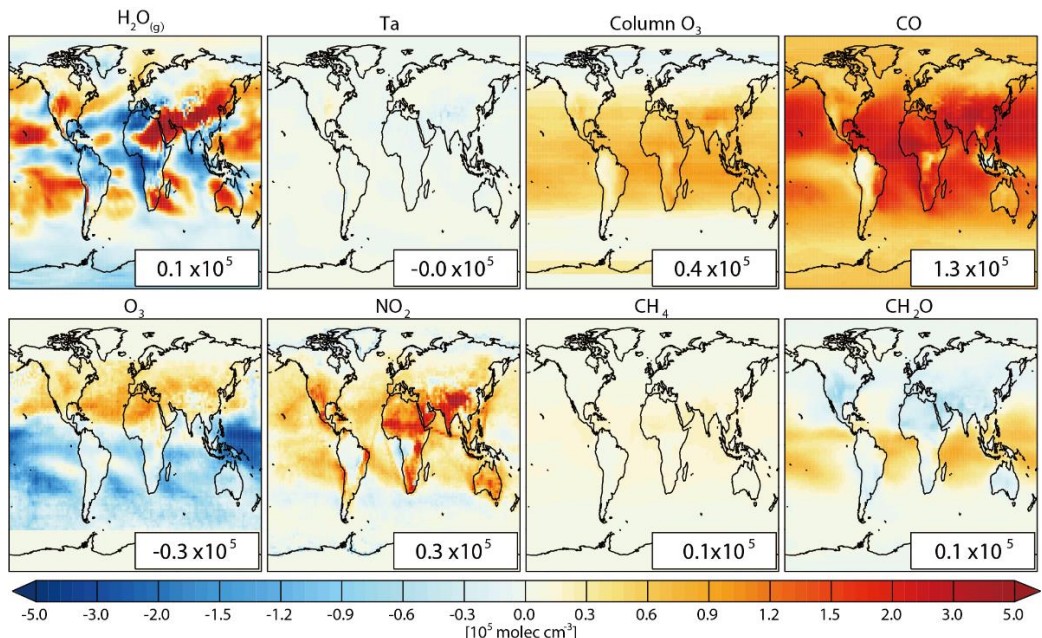

**Figure 3.** Spatial distributions of the contribution of individual factors to the difference between CESM1 CAM4-chem simulated and observation-based (modeled — observation-based) $[OH]_{trop-M}$. The global mean values are shown inset in molec cm$^{-3}$.



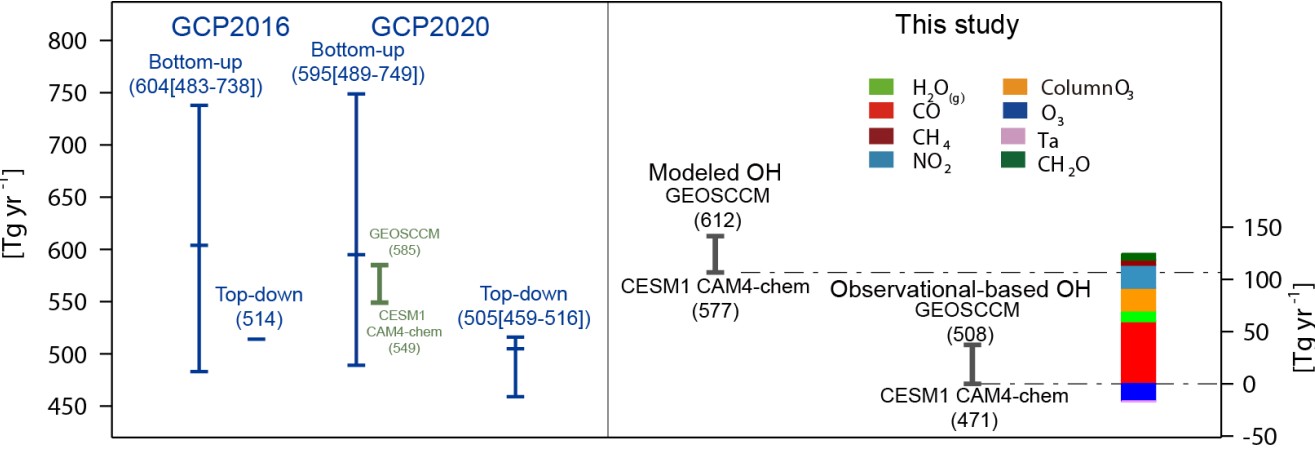

**Figure 4.** Global total chemical loss of CH$_4$ estimated by the bottom-up and top-down methods from previous GCP global CH$_4$ budget (blue bars; Saunois et al., 2016; 2020), simulated by GEOSCCM and CESM1 CAM4-Chem which is included in the bottom-up estimates in Saunois et al. (2020) (green bar), and that estimated in this study using the model simulated and observation-based OH fields and assimilated surface observations of CH$_4$ (black bars). The colored bar shows the contribution of individual factors to the difference in the chemical loss of CH$_4$ between CESM1 CAM4-Chem simulated and the corresponding observation-based OH. The blue, green, and black bars are corresponding to the left axis and the colored bar is corresponding to the right axis.

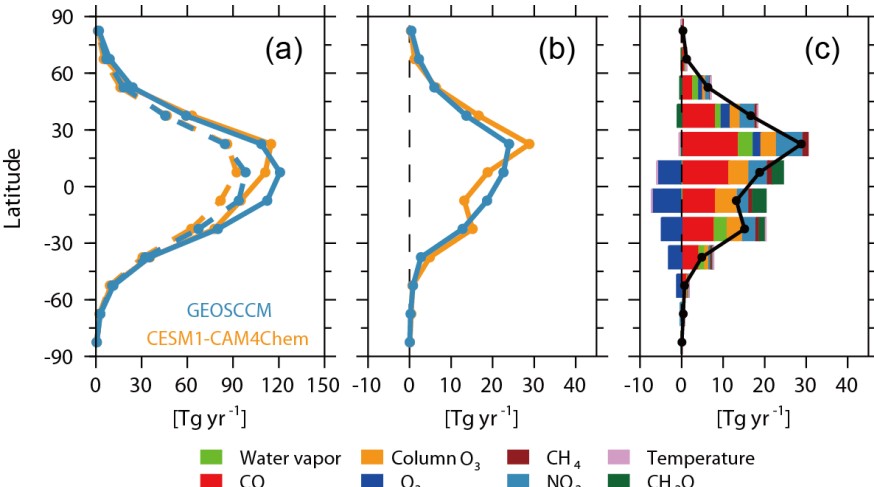

**Figure 5.** Same as Fig.2 but for the tropospheric CH$_4$ sink by reaction with OH.