# Peer review of "Reconciling the bottom-up and top-down estimates of the methane chemical sink using multiple observations"

_Atmospheric Chemistry and Physics, 2022_

## Author Comment (AC2)

**Reply to RC2: 'Comment on acp-2022-556'**

This paper aims to improve "bottom-up" estimates of OH concentrations by constraining chemical model simulations with observations of OH precursors. The paper is thorough, novel, well-written, and tackles a very important issue in atmospheric chemistry. I recommend it for publication in ACP, subject to some relatively minor corrections.

Response: We thank the reviewer for his/her helpful comments. All of them have been addressed in the revised manuscript. Please see our itemized responses below.

**General comments**

A simplified 0D model of atmospheric chemistry is used, gridcell-by-gridcell to determine the how the OH fields from a global 3D photochemical model would be adjusted by incorporating observations on OH precursors. One thing I felt was missing from the paper was a comparison of the OH fields predicted by the simplified model to that of the "parent" 3D model (i.e., how does [OH]\_DSMACC\_ref\_model compare to [OH]\_model, using the terms from eq. 1?). It seems that this is important because large differences could lead to non-linear effects that could influence the results. Perhaps some simple comparisons could be presented in the Supplement.

Response: We agree and compare the  $[OH]_{DSMACC\_Ref\_model}$  and  $[OH]_{model}$  in

the updated version of the text (L278-286): "The Ref\_model experiments can well reproduce the spatial distribution of  $[OH]_{trop-M}$  simulated by 3D models (Fig. S4), which indicate that the chemical box model DSMACC can generally capture the response of [OH] to the changes in OH precursor concentrations and meteorological conditions. However, the Ref\_model experiments overestimate the  $[OH]_{trop-M}$  by 7% and 36% when compared with the global  $[OH]_{trop-M}$  simulated by CESM1 CAM4-chem and GEOSCCM, respectively. Thus, the observation-based [OH] ( $[OH]_{obs}$ ) in each 3D model pixel for two different chemical mechanisms is estimated by correcting [OH] as simulated by the corresponding 3D models ( $[OH]_{model}$ ) by the ratio between [OH] simulated by DSMACC experiments for the All\_obs ( $[OH]_{DSMACC_all_obs}$ ) and for the Ref\_model

([OH]DSMACC\_Ref\_model) case"

We include Figure S4 in the supplement:

**Figure S4.** Spatial distributions of air mass-weighted tropospheric mean [OH] ( $[OH]_{trop-M}$ ) in 2010 from 3D model simulations (left) and chemical box model (DSMACC) simulations driven by the corresponding 3D model outputs (right). The global mean values are shown inset in molec cm-3.

**Specific comments**

**Comment 1:** On first reading, I was confused by the definition of the term: " $[OH]_{rop-M}$ ". When it is first introduced, both on line 44 and line 273, it is defined as a global value. E.g. "global tropospheric mean OH concentration" on line 44. However, it is later used to show regional distributions. I think that the authors are using  $[OH]_{trop-M}$  to mean something like "column average airmass-weighted [OH]", which is then sometimes averaged to produce a "global mean  $[OH]_{trop-M}$ "? I think the terminology needs to be tightened up a little here.

Response: We change "global tropospheric mean OH concentration" to "The global mean tropospheric column-averaged airmass-weighted [OH] ([OH]trop-M)" as suggested.

**Comment 2**: L83: "Such MCF-based top-down methods have..." rather than "method has".

Response: We change "method has" to "methods have" as suggested.

**Comment 3**: L105 – 107: I don't think these papers show that decreased [OH] can explain the resumed CH4 increase. Both have a high degree of uncertainty (such that no OH change is within the plausible range), and Rigby et al., 2017 has a coincident CH4 emissions increase in their maximum-likelihood estimate. I would perhaps keep it more general and say that these papers indicate that MCF-based top-down methods indicate that [OH] changes may have influenced recent CH4 trends, although with a

high degree of uncertainty.

Response: We change "conclude that decreased [OH] and therefore CH4 chemical loss after the mid-2000s can explain the resumed CH4 increase since 2006" to "indicate that the [OH] changes may have influenced recent CH4 trends, although with large uncertainties" as suggested.

**Comment 4**: L109: I don't think the models show a monotonic increase in [OH], do they? i.e, does the use of "continuous increase" need to be softened to "decadal trend" or similar?

Response: We change "continuous increase" to "positive decadal trend" as suggested.

L111 - 125: It seems that the Nicely et al. (2017; 2018) papers would fit into the discussion here too?

**Response: We add in the text:**

(L120-122) "Nicely et al. (2017; 2020) found that the inter-model difference in tropospheric [OH] is mainly driven by the difference in model simulated ultraviolet light flux to the troposphere, the tropospheric O3, CO, and NOx mixing ratio."

(L236-239) "Nicely et al. (2018) have estimated the response of [OH] to changes in OH precursors by conducting DSMACC model simulations for broad latitude and pressure bins. The results show that the  $H_2O_{(g)}$ ,  $NO_x$ , total column  $O_3$ , and tropical expansion can lead to a positive trend in tropospheric [OH], offsetting most of the negative trend led by the rising CH4 concentrations from 1980 to 2015."

We add reference:

"Nicely, J. M., Salawitch, R. J., Canty, T., Anderson, D. C., Arnold, S. R., Chipperfield, M. P., Emmons, L. K., Flemming, J., Huijnen, V., Kinnison, D. E., Lamarque, J.-F., Mao, J., Monks, S. A., Steenrod, S. D., Tilmes, S., and Turquety, S.: Quantifying the causes of differences in tropospheric OH within global models, Journal of Geophysical Research: Atmospheres, 122, 1983-2007, 10.1002/2016jd026239, 2017."

"Nicely, J. M., Canty, T. P., Manyin, M., Oman, L. D., Salawitch, R. J., Steenrod, S. D., Strahan, S. E., and Strode, S. A.: Changes in Global Tropospheric OH Expected as a Result of Climate Change Over the Last Several Decades, Journal of Geophysical Research: Atmospheres, 123, 10,774-710,795, doi:10.1029/2018JD028388, 2018."

"Nicely, J. M., Duncan, B. N., Hanisco, T. F., Wolfe, G. M., Salawitch, R. J., Deushi, M., Haslerud, A. S., Jöckel, P., Josse, B., Kinnison, D. E., Klekociuk, A., Manyin, M. E., Marécal, V., Morgenstern, O., Murray, L. T., Myhre, G., Oman, L. D., Pitari, G., Pozzer, A., Quaglia, I., Revell, L. E., Rozanov, E., Stenke, A., Stone, K., Strahan,

**S., Tilmes, S., Tost, H., Westervelt, D. M., and Zeng, G.: A machine learning examination of hydroxyl radical differences among model simulations for CCMI-1, Atmos. Chem. Phys., 20, 1341-1361, 10.5194/acp-20-1341-2020, 2020."**

**Comment 5:** L221: "... DSMCC is/was run forward" (insert is or was) **Response: We add "is" as suggested**

**Comment 6:** L223: "DAMSCC" should be changed to "DSMACC" L235: "observation-based", rather than "observational-based" **Response: We changed these typos as suggested.**

**Comment 7:** L239: I suggest "... [OH] simulated by DSMACC experiments for the All\_obs ( $[OH]_{DSMACC\_all\_obs}$ ) and for the Ref\_model ( $[OH]_{DSMACC\_Ref\_model}$ ) case" (add "case", or "simulation", or similar). **Response: We add "case" as suggested.**

L282 (and 315, 316 and 325): To improve readability, and given that it is only mentioned a couple of times, I suggest just referring to Spivakovsky et al. each time, rather than defining another term (S2000).

Response: We change "the S2000 OH field" to "the OH filed estimated by Spivakovsky et al. (2000) as suggested".

**Comment 8:** L300: should this by [OH]\_Trop-M, rather than [OH]? **Response: Yes, we change "[OH]" to "[OH]**trop-M".

L307: "which is larger than that over ..." (remove "the") **Response: we remove "the" in the text.**

**Comment 9:** L375: "over the 15 – 60N region" (insert "region" or similar) **Response: we add "region" as suggested.**

**Comment 10:** L384: "... by 0.07, but still cannot explain..." (insert "but") **Response: We add "but" as suggested.**

**Comment 11:** L395: "NO2 results in a positive bias" (insert "a") **Response: We add "a" as suggested.**

**Comment 12:** L404: Remove "The" from the start of the second sentence, or add "model" after "CESM1 CAM4-chem"

Response: We change "CESM1 CAM4-chem" to "CESM1 CAM4-chem simulation" as suggested.

**Comment 13**: L481: "... loss of CH4 in the previous GCP..." (add "the") L522: "... respectively, dominating the bias" (dominating, rather than

**"dominant") Response: Changed as suggested.**

**Comment 14:** Section 4 (Conclusions): This section could be made more concise and readable. I suggest thinking about the paragraph structure so that ideas are grouped together more concisely and start each paragraph with a sentence describing the main point of the paragraph (at present lots of paragraphs start with phrases like "In addition", or "Also", which don't help to orientate the reader).

Response: we summarize the uncertainties led by model chemical mechanisms and the one led by the availability of the observation data and model outputs (L632-692):

[revised manuscript text omitted]

Since we only have the tropospheric NO2 VCDs, another key factor that could influence the tropospheric OH burden but is unconstrained in this study is NO2 in the free troposphere. Although the NO2 mixing ratio is usually less than 1 ppbv in the free troposphere, the sensitivity of [OH] to NO2 can be very high in low NO2 regions. However, a potential model bias due to lightning NOx emissions, which had proven to contribute significantly to the upper-tropospheric [OH] burden (Murray et al., 2013; Turner et al., 2018), is not adjusted in our study. Satellite retrievals for upper tropospheric NO2 (e.g. Belmonte Rivas et al., 2015; Marais et al., 2021) could help quantify [OH] biases due to free tropospheric NO2 and the

**contribution of lightning NOx emissions. "**

**Comment 15:** L526: add "major" before "global CH4 sink", to make it clear that you're referring to one of the methane sinks (i.e., you're not also investigating Cl, etc.)

L586: "Such a difference is partly attributable to..." (remove "be")

L593: Remove "In addition"

L627: Remove "Also"

Response: Changed as suggested.

**Comment 16:** Figure 5: Consider making this a 2-panel plot (well, really a 6-panel plot) merged with Figure 2.

---

## Author Response (AR1)

**Reply to RC1: 'Comment on acp-2022-556'**

**Comment:** This is a very interesting study, presenting a relatively simple approach to correct 3D CTM generated tracer fields using a box model and satellite observed photochemical trace gases, leading to global mean OH estimates and interhemispheric OH differences that are closer to those derived from MCF. This is a very encouraging finding, suggesting that our understanding of photochemistry and methyl chloroform are good enough to allow a reduction in the uncertainty of OH. That is, if the two selected models are representative of that uncertainty, which is limited by n=2, meaning that the convergence between model and MCF derived constraints on OH might still arise from a fortunate coincidence. Nevertheless, the results look promising enough to investigate further.

The paper is very well written and with a logical story line and results that can rather easily be understood. In part this is due to the choice for a level of detail that keeps the focus on the main findings. This is good, however, some important details are missing that would be needed for someone to be able to repeat what was done. In addition, the validity of some assumptions should either be tested or discussed in further detail as explained below. With those issues addressed, which will at most call for moderate revisions, the paper should be acceptable for publication.

**Response:** We thank the reviewer for his/her helpful comments. All of them have been addressed in the revised manuscript. Please see our itemized responses below.

**Scientific comments**

**Comment 1:** Line 130-132: What is missing here is the use of chemical data assimilation, which is trying to achieve the same as this study, but through a more formal data assimilation procedure. A brief discussion is required of the relation between such methods and the method proposed here. The results should also be put in perspective of what has been achieved, or is achievable, using such methods.

**Response:** We now discuss the data assimilation method in the text (L131-139):

"The chemistry reanalysis that assimilates satellite observations of O3, CO, NO2, nitric acid (HNO3), and CO shows significant improvement on both global OH burden and inter-hemispheric gradient (Miyazaki et al., 2020). Such data assimilation methods can well balance the model and observation uncertainties, but they are not easy to apply to different models that simulate the broad range of global OH burden (Naik et al., 2013; Zhao et al., 2019). In addition, they do not allow partitioning the OH bias due to each precursor. In this context, the main objective of this study is to explore a simple approach to reconcile bottom-up and top-down estimates of the CH4 sink by (i) improving the simulated atmospheric OH fields using multiple satellite observations and meteorological data from reanalysis and (ii) assessing the contribution of each main OH precursor to the bias in simulated [OH] and CH4 sink."

We add reference: "Miyazaki, K., Bowman, K., Sekiya, T., Eskes, H., Boersma, F., Worden, H., Livesey, N., Payne, V. H., Sudo, K., Kanaya, Y., Takigawa, M., and

Ogochi, K.: Updated tropospheric chemistry reanalysis and emission estimates, TCR-2, for 2005–2018, Earth Syst. Sci. Data, 12, 2223-2259, 10.5194/essd-12-2223-2020, 2020."

**Comment 2:** Line 135: Why were CESM1-CAM4chem and GEOSCCM chosen from the CCMI-1 ensemble? What makes them representative members?

**Response:** We have clarified this point in the text (L208-213):

"We chose the CESM1 CAM4-Chem and GEOSCCM in this study since (1) their global mean OH concentrations and OH distributions (both horizontal and vertical) are around the multi-model mean values given by Zhao et al. (2019), albeit not at the extreme of the model distribution; (2) the two models include multiple primary non-methane volatile organic compounds (NMVOC) emissions (Morgenstern et al., 2017); (3) the chemical box model DSMACC already include the MOZART-4 and GEOS-Chem chemical mechanisms (section 2.3), which are similar to that used in CESM1 CAM4-Chem and GEOSCCM, respectively."

In this study, we do not aim to represent all the models that participate in the CCMI by the CESM1 CAM4-Chem and GEOSCCM. Instead, the models that simulate extreme OH distributions should also be tested in future work. We discuss in the text (L699-702):

"One future research development is to generate observation-based OH fields for all the atmospheric chemistry models included in the GCP global CH4 budget and over a longer time period, especially for the models that simulate extremely high or low [OH]. This will allow us to see if our results can be generalized with a larger range of [OH] and CH4 losses and to see if a higher consistency can also be achieved on longer timescales."

**Comment 3:** Line 150: Data availability is less relevant than the time window of the data that was actually used. Only towards the end it became clear that only the year 2010 was used for the observation-based box model calculations. Does that mean that only 2010 O3 data were used? This should be clear for other compounds also.

Response: To make it clear that we only use the observational data for 2010, we remove the time-window here. We also add in the text (L256-259): "In the All\_obs simulation, the CO, NO2, O3, CH4, and CH2O, total column O3, Ta, and H2O(g) are replaced with the available observation-based data for 2010, while other DSMACC inputs (pressure and other chemical species) are the same as in Ref\_model simulation."

**Comment 4:** Line 152: How is the troposphere defined in the model? How about the vertical O3 gradient within the troposphere in the application of the box model. Is the tropospheric mean applied to all tropospheric levels? Is there any use of averaging kernels? If not, how consistent is the observational adjustment of vertical profiles?

Response: The OMI/MLS datasets used in this study do not provide averaging kernel, so we cannot account for the vertical sensitivity of the satellite retrievals. Since satellite data only provide the tropospheric column density for O3, we estimate the observation-based O3 concentrations by combining the satellite-observed column density with the model-simulated vertical profiles. Here we cannot adjust the model-simulated ozone vertical distribution to observations using only satellite columns. We have clarified this point in the text(L262-273):

"For tropospheric NO2 and O3, we use satellite data to generate the observationbased DSMACC input. The associated uncertainties of using the satellite observations of O3 and NO2 at overpass time are discussed in section 4.3. As the satellite observations provide the tropospheric VCDs, the observation-based concentrations are estimated by combining the satellite observed tropospheric columns and model simulated vertical distributions. We estimate the tropospheric column density simulated by atmospheric chemistry models ( $C_{trop_model}$ ) using the tropopause pressure estimated based on the WMO tropopause definition (World Meteorological Organization, 1957). Then we estimate the scaling factor for each model horizontal grid cell as the ratio of satellite observations ( $C_{trop_obs}$ ) to the modeled tropospheric column density ( $C_{trop_model}$ ). The observation-based concentration ( $C_{grid_obs}$ ) in each model pixel, which is used as the DSMACC input, is then estimated by multiplying the corresponding model simulated concentration ( $C_{grid_model}$ ) by the scaling factor:

$$C_{grid\_obs} = C_{grid\_model} \times \frac{C_{trop\_obs}}{C_{trop\_model}}$$
(1)"

**Comment 5:** Line 164: How is the planetary boundary layer defined in the analysis? Since the sensitivity of the NO2 retrieval does not stop abruptly at the top of the PBL, to which altitudes is it applied and how is the sensitivity of satellite retrieved NO2 to the free troposphere accounted for?

Response: The boundary layer height is taken from MERRA-2 reanalysis data. As we explain in our response to comment 4, for NO2, we also estimate the scaling factor using the tropospheric vertical column density from satellite observations and model simulation for vertical distribution. But we only use this scaling factor to estimate the observation-based NO2 concentrations in the boundary layer. We have clarified this in the text (L274-276):

"For O3, we estimate the  $C_{grid\_obs}$  for each 3D model pixel in the whole troposphere using equation 1. For NO2, we only estimate  $C_{grid\_obs}$  in the boundary layer (the boundary layer height is from the MERRA-2 reanalysis data) since the NO2 emitted from the surface mainly remains within the boundary layer." **Comment 6:** Line 221: This assumes that the photochemistry is in diurnal steady state at the time when satellites measure the atmosphere. What supports this assumption? **Comment 7:** Line 231: Why are monthly means chosen if the satellite sampling is restricted to daytime satellite overpasses? How can these two be compared?

Response: We answer these two comments. Most of the CCMI models provide 3D outputs for chemical species with a monthly time resolution. This is a limitation we had to deal with. The original text was not precise enough on why and how we use the satellite data/observations and the 3D model output to drive the DSMACC model simulations.

We first clarify the diurnal cycle in OH concentrations estimated by the DSMACC model is driven by the diurnal cycle of photolysis rate estimated by the TUV model (L232-L235):

"Forced by meteorological variables  $(H_2O_{(g)}, T_a, and pressure)$ , total column O3, and gas concentrations simulated by the CESM1 CAM4-chem and GEOSCCM or generated from observations, and the diurnal cycle of the photolysis rates estimated by TUV radiation model, the DSMACC is run forward until reaching the diurnal steady state of OH."

We then clarify how we use the 3D model outputs to drive the reference experiment (L246-256):

"The reference experiment (Ref\_model in Table 1) is conducted by running the DSMACC model with the monthly mean chemical species concentrations and meteorological conditions simulated by the 3D models (CESM1 CAM4-chem/GEOSCCM) for each pixel in 2010 using the corresponding chemical mechanisms. During the DSMACC simulation for each month, the meteorological conditions and chemical species with lifetime from a few hours (e.g. NMVOCs) to several years (e.g. CH4) are set to the monthly mean values from 3D model outputs and unchanged during the simulation. We estimated the diurnal steady state solution for the chemical species with short lifetime of a few seconds (e.g. OH and HO2 radicals). Since most of the CCMI models provide the 3D distributions of the chemical species on monthly time resolution, the influence of sub-monthly variations such as the diurnal cycle for these chemical species and meteorological conditions are not represented in the DSMACC simulations."

As the 3D model outputs that are used to drive the reference experiments are on monthly time resolution, we also use the monthly mean data for most of the observations, as we have clarified in the text (L256-262):

"In the All\_obs simulation, the CO, NO2, O3, CH4, and CH2O, total column O3, Ta,

and  $H_2O_{(g)}$  are replaced with the available observation-based data for 2010, while other DSMACC inputs (pressure and other chemical species) are the same as in Ref\_model simulation. For CO, CH4, CH2O, and meteorological conditions, the observation-based data are directly taken from the monthly mean of the assimilated/reanalysis data as described in section 2.2 (regrid to model horizontal and vertical grid). For tropospheric NO2 and O3, we use satellite data to generate the observation-based DSMACC input. The associated uncertainties of using the satellite observations of O3 and NO2 at overpass time are discussed in section 4.3. As the satellite observations provide the tropospheric VCDs, the observationbased concentrations are estimated by combining the satellite observed tropospheric columns and model simulated vertical distributions."

Indeed, for tropospheric O3, of which the tropospheric mean lifetime is estimated as  $23.4 \pm 2.2$  days (Young et al., 2013), we assume that not considering the diurnal variation has a small impact. But for NO2, which has a shorter lifetime, we may overestimate the model high bias when comparing the monthly mean model output with satellite observations at the overpass day-time. Considering the complex factors that influence the NO2 diurnal variations, it is not easy to evaluate the uncertainties. We discuss the corresponding uncertainties in the text (L668-683):

"OMI measures concentrations of chemical species around local time 13:30, but most of the CCMI models only provide monthly means for 3D distribution of chemical concentrations. The monthly mean NO2 and O3 concentrations simulated by 3D models are therefore constrained only by such afternoon observations. For

O3, of which the tropospheric mean lifetime is 23.4±2.2 days (Young et al., 2013),

we assume that not considering diurnal variations has only a small influence. This is not the case for NO2 with a much shorter lifetime (~1 day, Jaffe et al., 2003). By comparing the tropospheric NO2 VCDs observed by SCIAMACHY (SCanning Imaging Absorption SpectroMeter for Atmospheric Chartography; overpass time around local time 10:00) with OMI, previous studies show that the tropospheric NO2 VCDs have significant diurnal variations (Boersma et al., 2008; 2009). Diurnal variations of NO2 VCDs are controlled by complex factors including local emissions, photochemistry, deposition, advection, etc., and vary among different seasons over different regions (Boersma et al., 2008; 2009). Considering the diurnal cycle of NO2 photolysis, tropospheric NO2 VCDs over remote regions should be lower during daytime than nighttime (Cheng et al., 2019). Constraining the model simulated monthly mean NO2 VCDs with satellite data at the overpass time leads to an overestimation of the high bias of modeled tropospheric NO2 VCDs. Thus, the 0.3×105 molec cm-3 estimated in this study gives an upper limit of the high bias in global [OH]trop-M due to boundary layer NO2."

Reference

Young, P. J., Naik, V., Fiore, A. M., Gaudel, A., Guo, J., Lin, M. Y., Neu, J. L., Parrish, D. D., Rieder, H. E., Schnell, J. L., Tilmes, S., Wild, O., Zhang, L., Ziemke, J. R., Brandt, J., Delcloo, A., Doherty, R. M., Geels, C., Hegglin, M. I., Hu, L., Im, U., Kumar, R., Luhar, A., Murray, L., Plummer, D., Rodriguez, J., Saiz-Lopez, A., Schultz, M. G., Woodhouse, M. T., and Zeng, G.: Tropospheric Ozone Assessment Report: Assessment of global-scale model performance for global and regional ozone distributions, variability, and trends, Elementa, 6, 10, https://doi.org/10.1525/elementa.265, 2018.

Boersma, K. F., Jacob, D. J., Trainic, M., Rudich, Y., DeSmedt, I., Dirksen, R., and Eskes, H. J.: Validation of urban NO2 concentrations and their diurnal and seasonal variations observed from the SCIAMACHY and OMI sensors using in situ surface measurements in Israeli cities, Atmos. Chem. Phys., 9, 3867-3879, 10.5194/acp-9-3867-2009, 2009.

Boersma, K. F., Jacob, D. J., Eskes, H. J., Pinder, R. W., Wang, J., and van der A, R. J.: Intercomparison of SCIAMACHY and OMI tropospheric NO2 columns: Observing the diurnal evolution of chemistry and emissions from space, 113, https://doi.org/10.1029/2007JD008816, 2008.

Cheng, S., Ma, J., Cheng, W., Yan, P., Zhou, H., Zhou, L., and Yang, P.: Tropospheric NO2 vertical column densities retrieved from ground-based MAX-DOAS measurements at Shangdianzi regional atmospheric background station in China, Journal of Environmental Sciences, 80, 186-196, https://doi.org/10.1016/j.jes.2018.12.012, 2019.

Jaffe, D.: Nitrogen Cycle, Atmospheric, in: Encyclopedia of Physical Science and Technology (Third Edition), edited by: Meyers, R. A., Academic Press, New York, 431-440, https://doi.org/10.1016/B0-12-227410-5/00922-4, 2003.

**Comment 8:**

(1) Line 235: How are satellite data that represent sub-column averages with variable vertical sensitivities regridded in the vertical?

**Response: As we have shown in our response to comment 4, we estimate the observation-based concentrations by combining the satellite-observed column density with the model-simulated vertical profiles (in main text L261-273).**

(2) What happens if the set of observations that is imposed to the box model (as I understand it) is inconsistent with the photochemistry scheme? Is there some nudging involved, or how do you prevent that non-observed compounds do not end up in an unstable solution?

**Response:** We have clarified how the DSMACC model works in the text (L249-253):

"During the DSMACC simulation for each month, the meteorological conditions and chemical species with lifetime from a few hours (e.g., NMVOCs) to several years (e.g. CH4) are set to the monthly mean values from 3D model outputs and unchanged during the simulation. We estimated the diurnal steady state solution for the chemical species with short lifetime of a few seconds (e.g., OH and HO2 radicals)."

Here, we can treat the chemical compounds input to the DSMACC model (either from observations or from 3D model simulations) as "external forcing", and the DSMACC model estimates the steady-state concentrations of OH and other short-lifetime chemical species (e.g., HO2) under the given meteorological conditions, long lifetime species concentrations, and the photolysis rates estimated by the TUV model. Since it is easy to reach the steady state for these short-lifetime chemical species, all simulations can finally reach the steady state solution.

**Comment 9:** Equation 2: This equation assumes that the full 3D OH\_model for 2010 that is supposed to be represented by OH\_DSMACC\_REF\_MODEL indeed match each other on the monthly mean time scale for 2010. I did not find any evidence that this is the case, or the extent to which this requirement is satisfied.

Response: To make the DSMACC model simulation more consistent with the 3D model simulations, we compile the DSMACC model with chemical mechanisms that are similar to the corresponding 3D model simulations(L229-231):

"In this study, the DSMACC model is compiled with MOZART-4 and GEOS-Chem chemical mechanisms, respectively, to be consistent with the associated 3D models CESM1 CAM4-chem and GEOSCCM."

And the  $[OH]_{DSMACC\_Ref\_model}$  is estimated by inputting the monthly mean concentrations and meteorological conditions simulated by 3D models (L246-249):

"The reference experiment (Ref\_model in Table 1) is conducted by running the DSMACC model with the monthly mean chemical species concentrations and meteorological conditions simulated by the 3D models (CESM1 CAM4-chem/GEOSCCM) for each pixel in 2010 using the corresponding chemical mechanisms."

We compare the  $[OH]_{DSMACC\_Ref\_model}$  and  $[OH]_{model}$  in the text (L278-282):

"The Ref\_model experiments can well reproduce the spatial distribution of [OH]trop-M simulated by 3D models (Fig. S4), which indicate that the chemical

box model DSMACC can generally capture the response of [OH] to the changes in OH precursor concentrations and meteorological conditions. However, the Ref\_model experiments overestimate the  $[OH]_{trop-M}$  by 7% and 36% when compared with the global  $[OH]_{trop-M}$  simulated by CESM1 CAM4-chem and GEOSCCM, respectively. Thus, the observation-based [OH] ( $[OH]_{obs}$ ) in each 3D model pixel for two different chemical mechanisms is estimated by correcting [OH] as simulated by the corresponding 3D models ( $[OH]_{model}$ ) by the ratio between [OH] simulated by DSMACC experiments for the All\_obs

([OH]DSMACC\_all\_obs) and for the Ref\_model ([OH]DSMACC\_Ref\_model) case : "

**Figure S4.** Spatial distributions of air mass-weighted tropospheric mean [OH] ([OH]trop-M) in 2010 from 3D model simulations (left) and chemical box model (DSMACC) simulations driven by the corresponding 3D model outputs (right). The global mean values are shown inset in molec cm-3.

**Here the $[OH]_{DSMACC\_Ref\_model}$ is not exactly the same as the [OH] simulated by**

the 3D model  $([OH]_{model})$ , but we think  $\frac{[OH]_{DSMACC\_all\_obs}}{[OH]_{DSMACC\_Ref\_model}}$  reflects the relative changes in [OH] when the meteorological conditions and OH precursor concentrations are constrained by observations.

**Comment 10:** Line 258: Does 'I' run over the troposphere or the entire atmosphere? Equation 4 suggests the troposphere, but equation 7 the whole atmosphere (for the global CH4 burden). This should be clarified.

Response: We clarify in the text (L304-305; L315-316):

$$\delta L_{CH4+OH_xk} = \sum_i \sum_t K(T) m(CH_4) \delta[OH]_{xk} \delta t$$
(4)

Where i is the index of the model pixel in the troposphere and  $\delta t$  is the integration time step (3 hours).

$$\tau_{CH4+OH} = \frac{\sum_{i} m(CH_4)}{L_{CH4+OH}}$$
(7)

**Where j is the index of the model pixel in the entire atmosphere.**

**Comment 11:** Line 283: In the TRANSCOM-CH4 experiment a scaling factor of 0.92 was applied to the Spivakovsky fields based on a MCF inversion by Krol et al.

Response: we add in the text (L327-332): "The global [OH]trop-M estimated by the observation-based OH fields in this study is lower than the value estimated by Spivakovsky et al. (2000) (11.6×105 molec cm-3), which is used in the chemistry-transport model (CTM) intercomparison experiment (TransCom-CH4) after scaled by a factor of 0.92 (Patra et al., 2011), but consistent with those estimated by MCF-based inversions (~10×105 molec cm-3; Bousquet et al., 2005; Krol and Lelieveld, 2003)."

**Comment 12:** Figure 1: How realistic are the OH holes over tropical rainforests given what is known about radical recycling under low NOx conditions?

Response: As we discussed in the text, the method presented in this study cannot reduce the uncertainties led by different chemical mechanisms. We emphasize the uncertainty led by not including the OH recycling by isoprene in the CESM1-CAM4chem and GEOSCCM chemical mechanism (L650-654):

"Both CESM1-CAM4chem and GEOSCCM do not include the OH recycling by isoprene and simulate low OH values in regions with high NMVOC emissions, such as rain forests in the Southern Hemisphere (Zhao et al., 2019). Including the chemical mechanism such as OH recycling by isoprene (Lelieveld et al. 2008) would help further reduce the N/S ratio for model-simulated OH fields."

**Comment 13:** Line 341: This is a surprising finding, especially since there must be correlated regional adjusments in for e.g NOx and CO. The reason could be that the adjustments are small enough. The size of regional adjustments is not shown, but could be quite substantial. The statement that the non-linearity of photochemistry is negligible globally should be backed up by a test that it is significant regionally, which we know it is. If it is not, then I wonder what is going wrong.

**Response:** We compare the adjustments on the regional scale in Fig.S8. They usually represent

**Figure S8** (a) Zonal averaged difference between modeled and observation-based  $[OH]_{trop-M}$  estimated by the All\_obs simulation  $([OH]_{model} - [OH]_{obs};$  yellow); The total contribution of the 8 individual factors to the difference in global  $[OH]_{trop-M}$  estimated from the simulation xk\_obs simulations  $(\sum \delta [OH]_{xk};$  blue). (b) The difference between the two estimates  $(\sum \delta [OH]_{xk} - ([OH]_{model} - [OH]_{obs}))$ . We have changed the text (L386-387) as:

"On the global scale, the total contribution of the 8 individual factors to the difference in  $[OH]_{trop-M}$  estimated from the simulation xk\_obs is  $2.0 \times 10^5$  molec cm-3 (Table 4), consistent with that estimated from the simulation All\_obs (Table 2). On the regional scale, they show small differences (usually <10% of the signal, Fig. S8), which can be attributed to the nonlinear chemistry. Indeed, although the atmospheric OH is produced and removed through complex nonlinear chemical reactions, one can infer the large-scale [OH]\_trop-M changes by roughly summing the influence from individual factors."

**Comment 14**: Line 486: Here the reader should be reminded that this holds for the period 2000-2009.

**Response:** We add the time period in the text (L525-527):

"After adjusting the main OH precursors to observations, the global chemical sink of CH4 for 2000-2009 is 471-508 Tg yr-1, as estimated using the two observationbased OH fields, more consistent with top-down method estimates (~500Tg yr-1)."

**Technical corrections**

Line 390: "northern China" i.o. "North China" Line 420: "the" i.o. "such as" Line 453: "limited' i.o. "a few" Line 481: "in the previous" i.o. "in previous" Line 541: "molec cm-3" i.o. "moelc cm-3" Line 545: "krol" i.o. "korl" **Response: The technical corrections are revised as suggested.**

**Reply to RC2: 'Comment on acp-2022-556'**

This paper aims to improve "bottom-up" estimates of OH concentrations by constraining chemical model simulations with observations of OH precursors. The paper is thorough, novel, well-written, and tackles a very important issue in atmospheric chemistry. I recommend it for publication in ACP, subject to some relatively minor corrections.

Response: We thank the reviewer for his/her helpful comments. All of them have been addressed in the revised manuscript. Please see our itemized responses below.

**General comments**

A simplified 0D model of atmospheric chemistry is used, gridcell-by-gridcell to determine the how the OH fields from a global 3D photochemical model would be adjusted by incorporating observations on OH precursors. One thing I felt was missing from the paper was a comparison of the OH fields predicted by the simplifi